# A Prompt-Based Knowledge Graph Foundation Model for Universal In-Context Reasoning

**Yuanning Cui[†], Zequn Sun[†], Wei Hu[†‡∗]**
[†]State Key Laboratory for Novel Software Technology, Nanjing University, Nanjing, China
[‡]National Institute of Healthcare Data Science, Nanjing University, Nanjing, China
yncui.nju@gmail.com, {sunzq, whu}@nju.edu.cn

## Abstract

Extensive knowledge graphs (KGs) have been constructed to facilitate knowledge-driven tasks across various scenarios. However, existing work usually develops separate reasoning models for different KGs, lacking the ability to generalize and transfer knowledge across diverse KGs and reasoning settings. In this paper, we propose a prompt-based KG foundation model via in-context learning, namely **KG-ICL**, to achieve a universal reasoning ability. Specifically, we introduce a prompt graph centered with a query-related example fact as context to understand the query relation. To encode prompt graphs with the generalization ability to unseen entities and relations in queries, we first propose a unified tokenizer that maps entities and relations in prompt graphs to predefined tokens. Then, we propose two message passing neural networks to perform prompt encoding and KG reasoning, respectively. We conduct evaluation on 43 different KGs in both transductive and inductive settings. Results indicate that the proposed KG-ICL outperforms baselines on most datasets, showcasing its outstanding generalization and universal reasoning capabilities. The source code is accessible on GitHub: https://github.com/nju-websoft/KG-ICL.

## 1 Introduction

Reasoning on knowledge graphs (KGs) involves inferring new relational facts from existing ones. Early related work primarily focuses on reasoning over a static KG in the transductive setting, but lacks the generalization ability to handle new entities or relations in the KG. Recent research [1, 2, 3, 4] considers the relational patterns between seen and unseen entities, enabling inductive reasoning. However, these methods still lack the transferability to reason over unseen KGs due to the unshared and unlinked entity and relation vocabularies between the pre-trained KG and unseen KGs.

The primary challenge in generalizing to new entities, relations, and even different KGs lies in how to represent such unseen data. Some methods [1, 2, 3, 4] aggregate query-conditioned relational structures to represent entities. They can conduct inductive reasoning over unseen entities using these relative entity representations without the need of pre-trained entity embeddings. However, these methods cannot reason over unseen relations. To resolve this issue, some recent methods [5, 6] develop relative relation representations. They model relation interactions using a query-conditioned relation graph, where each node represents a relation and an edge indicates that the linked two relations share a subject or object entity in the KG. They conduct message passing on the query-conditioned relation graph to represent relations.

However, the relation graph only describes the connectivity of relations in the KG, with less attention to the local context of the entity and relation in a query. As a result, these methods usually fail to

---

[∗]Corresponding author

38th Conference on Neural Information Processing Systems (NeurIPS 2024).

generate discriminative relation representations. For example, to infer the query relation `parentOf`, the most relevant relation is `coupleOf`. While in the KG, since every student has parents and most teachers are parents, the relation graph would also contain edges "`parentOf` ⇒ `teach`" and "`teach` ⇒ `parentOf`". The relation `teach` appears as noise in representing `parentOf`, which may mislead the model, resulting in prediction failures. This inspires us to capture the local contexts and highlight the important relations relevant to queries, rather than relying on a global relation graph.

In this paper, we propose a novel KG reasoning foundation model with in-context learning, namely KG-ICL. In-context learning is a method that allows pre-trained models to learn tasks based on only a few examples without updating model parameters. The extraordinary success of in-context learning in language modeling [7] hinges on three crucial fundamentals: prompt design, unified tokenization, as well as contextual understanding and utilization.

The art of prompt design lies in highlighting task-critical information. We construct a prompt graph to model query-related contexts, which starts with an example fact about the query relation, i.e., (`subject`, `query relation`, `object`). We consider two types of contexts as prompts. The first is entity context, which includes the neighboring entities of the example subject and object. The second is relation context, which considers relational paths between the subject and object entities. Thus, the node set of our prompt graph includes the neighbors of the example subject and object, as well as the entities within the paths connecting the subject and object in the KG. We utilize the induced subgraph of these entities as a prompt graph.

Then, we design a unified tokenizer that is applicable to various prompt graphs. The key challenge is that the entities and relations usually vary across different KGs [8, 9], and this issue extends to prompt graphs as well. Conventional KG reasoning models [10, 11, 12, 13, 14] merely learn an individual embedding for each entity or relation, resulting in the inability to reason over unseen KGs. We extend the entity labeling method of GraIL [1] to relations, proposing a unified tokenizer for various prompt graphs. Given a query relation and its prompt graph, we first group the involved entities based on the lengths of their shortest path to the example subject and object entities. Similarly, we categorize relations into two classes depending on whether they represent query relations. Finally, the entities or relations in the same group will be mapped to the same token. As a result, prompt graphs from different KGs are described in "the same language".

Given the above prompt graph and unified tokenizer, we propose two message passing neural networks as the prompt encoder and KG reasoner, respectively. The input of the prompt encoder is the prompt graph and the learnable token representations. At each layer of prompt encoding, we introduce an entity-centric and a relation-centric aggregation. Notably, in relation-centric aggregation, we treat relations as special nodes and update their representations by aggregating messages from facts containing them. After prompt encoding, we read the relation representations from the prompt graphs to support KG encoding. At the beginning of KG encoding, we initialize the relation representations in the KG as the prompt relation representations. As for entities, we initialize the subject entity as the query relation representation, and other entities are initialized as zero vectors. After performing message passing over the KG, we score all entities based on the output entity representations.

We conduct extensive experiments on 43 datasets to validate the effectiveness of our model. The experimental results indicate that our model not only possesses universal reasoning capabilities across diverse KGs but also outperforms supervised and pre-training models. Moreover, we observe that the proposed model exhibits robustness and high efficiency in utilizing examples.

In summary, our main contributions are listed below:

- Our key contribution is an in-context KG reasoning foundation model. It prompts the pre-trained model to engage in relational reasoning over diverse KGs.

- We propose a prompt graph as context to support in-context learning. It consists of an example fact about the query relation and its relevant subgraphs and paths. We also employ a unified tokenizer to map entities and relations in prompt graphs to predefined tokens.

- Given a prompt graph with token representations, we propose two message passing networks for prompt graph encoding and KG reasoning. The foundation model can be further finetuned on specific KGs to obtain improved performance.

- We conduct extensive experiments on 43 KGs in both transductive and inductive settings to demonstrate the universal reasoning capability of our model.

## 2 Related Work

**KG reasoning.** KG reasoning primarily involves three settings: transductive, inductive, and fully-inductive. Early studies [10, 11, 12, 13, 14] focus mainly on the transductive setting, assuming that KGs are static. Real-world KGs are dynamic, inspiring the development of inductive models [1, 2, 3, 4, 15, 16, 17, 18, 19, 20, 21, 22, 23] that allows for emerging entities. In the fully-inductive setting [5, 24, 25, 26], both unseen entities and relations can emerge in the query facts. This setting remains limited to the same KG. In contrast, our in-context learning and KG foundation model seek to break down the barriers imposed by these settings and achieve universal reasoning capabilities.

**Prompt and in-context learning in graph pre-training.** Our work is also related to graph prompt learning and graph in-context learning. Inspired by the success of pre-training models in NLP [27] and computer vision [28], some graph pre-training models [29, 30, 31, 32, 33] have been proposed. These models follow the paradigm of "pre-train and finetune", where a model is initially pre-trained and then finetuned for the target task. The work [34] further develops a KG pre-training model. Consequently, recent work [8, 35, 36, 37, 38, 39, 40, 41, 42, 43, 44, 45] has shifted focus to the "pre-train, prompt, and finetune" paradigm. The relation graph of the KG pre-training model [6] can also be seen as a special prompt. This paradigm leverages task prompts to enhance the knowledge transfer and generalization abilities of pre-trained models. Inspired by the recent success of large language models like GPT [7], recent work uses in-context learning to avoid finetuning. It imparts general capabilities to pre-trained models with just a few examples. PRODIGY [46] introduces an in-context learning-based model to handle various classification tasks on graphs. While it can perform relation classification, it is not suitable for KG reasoning with a massive number of candidate entities.

We discuss more related work in Appendix D.

## 3 Problem Definition

**KG Reasoning.** We define a KG as $\mathcal{K} = (\mathcal{E}, \mathcal{R}, \mathcal{T})$, where $\mathcal{E}$, $\mathcal{R}$, and $\mathcal{T}$ denote the sets of entities, relations, and facts, respectively. A fact $(s, r, o) \in \mathcal{T}$ consists of a subject entity $s \in \mathcal{E}$, a relation $r \in \mathcal{R}$, and an object entity $o \in \mathcal{E}$. Given a KG and a query fact in the form of $(s, q, ?)$, the reasoning task is to predict the missing entity from $\mathcal{E}$. We refer to the relation $q$ as a query relation.

In practice, we follow the convention [10] to introduce inverse relations. For each relation $r \in \mathcal{R}$, we add its inverse relation $r^-$ into the relation set and add the reverse fact $(o, r^-, s)$ into the fact set.

**In-Context KG Reasoning.** In in-context reasoning, a model is pre-trained using a set of source KGs, denoted by $\{\mathcal{K}_1, \ldots, \mathcal{K}_n\}$. After pre-training, the model conducts reasoning on emerging KGs based on only a few related examples without updating model parameters. Each pre-training or reasoning query is prompted with some relevant examples as context.

The prompt is crucial for in-context learning. For each query relation $q$, we first randomly sample some of its facts, e.g., $c = (u, q, v) \in \mathcal{T}$. Next, we extract a subgraph $\mathcal{P}_c = (\mathcal{E}_{\text{pmt}}, \mathcal{R}_{\text{pmt}}, \mathcal{T}_{\text{pmt}})$ from the KG for each example fact to construct a prompt graph. In the following, we provide a broad definition of prompt graphs, allowing for a broad design space:

**Prompt Graph.** Given an example fact $c = (u, q, v)$ in a KG $\mathcal{K} = (\mathcal{E}, \mathcal{R}, \mathcal{T})$, where $c \in \mathcal{T}$, we define its prompt graph $\mathcal{P}_c = (\mathcal{E}_{\text{pmt}} \subseteq \mathcal{E}, \mathcal{R}_{\text{pmt}} \subseteq \mathcal{R}, \mathcal{T}_{\text{pmt}} \subseteq \mathcal{T})$ as a subgraph of $\mathcal{K}$, and $c \in \mathcal{T}_{\text{pmt}}$.

To encode prompt graphs, we extend the KG-independent entity labeling [1] to relations and propose a unified tokenizer, which maps entities and relations from different KGs to unified tokens:

**Unified Tokenizer.** The unified tokenizer is a many-to-one mapping function. It maps entities and relations of different prompt graphs to the predefined tokens. Specifically, it maps each entity based on the length of its shortest paths to the subject and object entities of the example fact, i.e., $\text{tokenize}(e) \leftarrow [\text{dist}(u, e), \text{dist}(v, e)]$, where $\text{dist}(\cdot)$ is the length of the shortest path between two entities. It maps each relation to the tokens by whether it is the same as the query relation. That is, $\text{tokenize}(r) \leftarrow [\text{same}(r, q)]$, where $\text{same}(r, q) = 1$ if $r$ is the same as $q$, otherwise $\text{same}(r, q) = 0$.

In Section 4.2, we assign a learnable representation for each token.

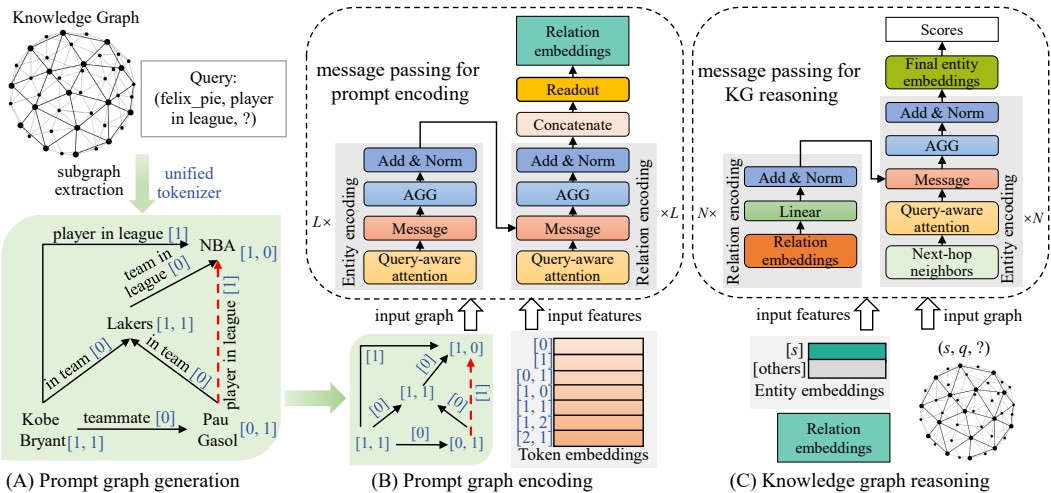

Figure 1: Overview of the in-context KG reasoning foundation model. (A) Given the query and KG, we extract prompt graphs as context for the query relation "player in league". The entities and relations in the prompt graphs are mapped to the unified tokens. (B) We employ a message passing neural network to encode the prompt graph and readout the relation representations as the prompts. (C) Then we use the prompts to initialize the representations of entities and relations in the KG. After KG encoding, we score the candidate entities according to their embeddings in the last layer.

## 4 In-context Reasoning over KGs

The overview of the proposed model is shown in Figure 1. Given a KG and a query, we first generate prompt graphs for the query relation. Then, we use an encoding module to encode the prompt graphs and readout prompts. Finally, we incorporate the prompts into the KG reasoning process.

### 4.1 Prompt Graph Generation

The prompt graph defined in Section 3 allows for a broad design space. In this section, we introduce a specific method for generating prompt graphs. We primarily address two challenges: (i) How to make the prompt graph general for diverse KGs? (ii) How to provide valuable prompts to enhance reasoning? We propose a prompt graph generation pipeline to address these challenges. It involves two steps: example sampling and prompt graph extraction.

**Example sampling.** For a query relation $q$, we first randomly sample $M$ example facts as follows:

$$\mathcal{S}_q = \{c_i\}_{i=1}^M, \quad c_i \sim \text{Uniform}(\mathcal{N}_q), \tag{1}$$

where $\mathcal{N}_q = \{(u, r, v) \,|\, r = q \wedge (u, r, v) \in \mathcal{T}\}$ and $c_i = (u, q, v)$ is a $q$-specific example fact.

**Prompt graph extraction.** The key point of the prompt graph design is highlighting information crucial for query relation-specific reasoning. The example fact consists of a subject entity, an object entity, and the query relation between them. To depict the example subject and object entities, we draw inspiration from the research on prompt-based graph model [35, 46] to use neighboring nodes centered around the central node to construct prompt graphs. To abstract the semantics of query relation, we include the paths between example subject and entities, considering the success of logical rules in KG reasoning [47, 48, 49, 50]. The body of the rules involves paths between the subject and object entities. Therefore, given an example fact $c = (u, q, v) \in \mathcal{S}_q$ and a KG $\mathcal{K} = \{\mathcal{E}, \mathcal{R}, \mathcal{T}\}$, we include the neighboring entities of $u$ and $v$ and the $k$-hop paths between $u$ and $v$ in the prompt graph:

$$\mathcal{E}_{\text{pmt}} = \{x \,|\, \exists (x, r, u) \in \mathcal{T}\} \cup \{x \,|\, \exists (x, r, v) \in \mathcal{T}\} \\ \cup \{x \,|\, \text{dist}(x, u) + \text{dist}(x, v) \le k\}, \tag{2}$$

where $k$ is a hyperparameter denoting the maximum value of $\text{dist}(x, u) + \text{dist}(x, v)$. As we have added reverse facts, $\mathcal{E}_{\text{pmt}}$ includes all 1-hop neighbors. Next, we extract the facts and relations among them, i.e., $\mathcal{T}_{\text{pmt}} = \{(s, r, o) \,|\, s \in \mathcal{E}_{\text{pmt}} \wedge o \in \mathcal{E}_{\text{pmt}} \wedge (s, r, o) \in \mathcal{T}\}$ and $\mathcal{R}_{\text{pmt}} = \{r \,|\, \exists (s, r, o) \in \mathcal{T}_{\text{pmt}}\}$.

## 4.2 Prompt Encoding

In this section, we design a message passing neural network for prompt encoding. It comprises three sub-modules: token representation, message passing, and readout. We begin by initializing the token representations of entities and relations in the given prompt graph. Subsequently, a multi-layer message passing neural network is employed to encode the prompt graph. Finally, we introduce a readout sub-module to obtain the prompt representation.

**Token representations.** We assign each token a learnable vector representation. Specifically, according to Equation (2), the tokens for entities satisfy $i + j \leq k$, $0 \leq i \leq k - 1$ and $0 \leq j \leq k - 1$. Therefore, we set a representation matrix $\mathbf{T} \in \mathbb{R}^{(\frac{(k+1)(k+2)}{2} - 2(k-1)) \times d}$ for entity tokens, where $\frac{(k+1)(k+2)}{2} - 2(k - 1)$ denotes the total number of entity tokens. As for relations, the representation of token $[z]$ is initialized as $\mathbf{q}^{\text{token}} \cdot z$, where $\mathbf{q}^{\text{token}} \in \mathbb{R}^{1 \times d}$ is a learnable representation. We denote the input representation matrix of entities and relations for the prompt graph as $\mathbf{H}_{\text{E}}^{(0)}$ and $\mathbf{H}_{\text{R}}^{(0)}$, respectively.

**Message passing for prompt graph.** Then, we employ an $L$-layers message passing neural network, which incorporates two types of aggregation: an entity-centric aggregation and a relation-centric aggregation. In each layer, we first update the entity representations as follows:

$$\mathbf{H}_{\text{E}}^{(l+1)} \leftarrow \underset{\forall e \in \mathcal{E}_{\text{pmt}}, \forall n \in \mathcal{N}_e}{\text{Aggregation}_{\text{E}}} \left( \left\{ \text{Message}(\mathbf{H}_{\text{E}}^{(l)}, \mathbf{H}_{\text{R}}^{(l)}, n, q) \right\} \right), \tag{3}$$

where $\mathcal{N}_e \subseteq \mathcal{T}_{\text{pmt}}$ is the set of facts containing the entity $e$, and $q$ is the query relation of this prompt graph. Then we update the relation representations using the updated entity representations and the relation representations from the previous layer:

$$\mathbf{H}_{\text{R}}^{(l+1)} \leftarrow \underset{\forall r \in \mathcal{R}_{\text{pmt}}, \forall n \in \mathcal{N}_r}{\text{Aggregation}_{\text{R}}} \left( \left\{ \text{Message}(\mathbf{H}_{\text{E}}^{(l+1)}, \mathbf{H}_{\text{R}}^{(l)}, n, q) \right\} \right), \tag{4}$$

where $\mathcal{N}_r \subseteq \mathcal{T}_{\text{pmt}}$ is the set of facts containing the relation $r$. Under this message passing framework, we present two specific aggregation and message functions in Appendix A.1.

**Readout.** After $L$-layers message passing on the prompt graph $\mathcal{P}$, we obtain the prompt as follows:

$$\mathbf{H}_{\mathcal{P}} = \mathbf{W}_{\text{Readout}} \left( \mathbf{H}_{\text{R}}^{(1)} \| \mathbf{H}_{\text{R}}^{(2)} \| \cdots \| \mathbf{H}_{\text{R}}^{(L)} \right), \tag{5}$$

where $\mathbf{W}_{\text{Readout}} \in \mathbb{R}^{d \times Ld}$ is a learnable weight matrix. Note that the relations in different prompt graphs may vary. We fill in the relations not present in the prompt graph with zero vectors to obtain $\hat{\mathbf{H}}_{\mathcal{P}} \in \mathbb{R}^{|\mathcal{R}| \times d}$, ensuring that the shapes of every representation matrix are the same. Finally, we use mean-pooling to aggregate the information from multiple prompt graphs as follows:

$$\overline{\mathbf{H}}_{\text{pmt}} = \frac{1}{|\mathcal{S}_q|} \sum_{c \in \mathcal{S}_q} \hat{\mathbf{H}}_{\mathcal{P}_c}, \tag{6}$$

where $\overline{\mathbf{H}}_{\text{pmt}} \in \mathbb{R}^{|\mathcal{R}| \times d}$ is the prompt relation representation matrix, $\mathcal{S}_q$ is the set of example facts of the query relation $q$, and $\mathcal{P}_c$ is the prompt graph corresponding to the example fact $c$. In practice, we parallel encode these prompt graphs to ensure efficiency.

## 4.3 In-Context KG Encoding and Reasoning

Based on the prompt encoding, we conduct reasoning on KGs. To achieve a KG-independent encoding, we draw inspiration from the conditional message passing neural network [3, 4, 20, 21, 22] to present a novel KG reasoning module. It separately encodes entities based on the query, rather than mapping them to specific embeddings, offering us an opportunity for knowledge transfer across diverse KGs. It comprises three sub-modules: initialization, KG encoding, and reasoning.

**Initialization.** The input relation representations in the KG are initialized as the prompt relation embeddings, i.e., $\mathbf{V}_{\text{R}}^{(0)} = \overline{\mathbf{H}}_{\text{pmt}}$. As for entity representations, given a query fact $(s, q, x)$, the representation of $s$ is initialized as the representation of the query relation, i.e., $\mathbf{s} = \mathbf{q}$. Other entities are represented by zero vectors. We denote the input representation matrix of entities in KG as $\mathbf{V}_{\text{E}}^{(0)}$.

**Message passing for KG.** Here we employ an $N$-layers message passing neural network to aggregate multi-hop information. At each layer, we first update relation representations as follows:

$$\mathbf{V}_{\mathrm{R}}^{(l+1)} = \mathrm{LN}\Big(\mathbf{V}_{\mathrm{R}}^{(l)} + \mathrm{ReLU}\big(\mathbf{W}_{\mathrm{R}}^{(l)}\mathbf{V}_{\mathrm{R}}^{(l)}\big)\Big), \tag{7}$$

where LN denotes the layer normalization operation, and $\mathbf{W}_{\mathrm{R}}^{(l)} \in \mathbb{R}^{d \times d}$ is a learnable weight matrix.

Then, we update entity representations based on the updated relation representations. Some studies [3, 20] have shown that the distance-based inductive bias is crucial for KG reasoning. Inspired by this, we introduce a hop-by-hop message passing neural network to update entity representations, starting from the subject entity and expanding one-hop neighbors at each layer:

$$\mathbf{V}_{\mathrm{E}}^{(l+1)} \leftarrow \underset{\forall e \in \mathcal{L}^{(l+1)}, \forall n \in \mathcal{N}_e}{\mathrm{Aggregation}_{\mathrm{E}}} \Big(\big\{\mathrm{Message}(\mathbf{V}_{\mathrm{E}}^{(l)}, \mathbf{V}_{\mathrm{R}}^{(l+1)}, n, q)\big\}\Big), \tag{8}$$

where $q$ is the query relation, $\mathcal{L}^{(l)}$ is the set of entities in $l$-hop neighbors of $s$, and $\mathcal{L}^{(0)} = \{s\}$, $\mathcal{L}^{(l+1)} = \mathcal{L}^{(l)} \cup \big\{e \mid \exists (x, y, e) \in \mathcal{T} \wedge x \in \mathcal{L}^{(l)}\big\}$. Under this message passing framework, we present a specific message passing neural network for KG encoding in Appendix A.2.

**Reasoning.** Finally, we read the representations of candidate entities and assign scores to them, i.e., $f(s, q, e) = \mathbf{W}_{\mathrm{score}}\mathbf{e}_{s,q}^{(N)}$, where $\mathbf{e}_{s,q}^{(N)} \in \mathbf{V}_{\mathrm{E}}^{(N)}$ is the output representation of the entity $e$, and $\mathbf{W}_{\mathrm{score}} \in \mathbb{R}^{1 \times d}$ is a weight matrix. Note that the message passing neural network we employ for KG encoding is conditioned on specific queries of the form $(s, q, ?)$, meaning the output representations $\mathbf{V}_{\mathrm{E}}^{(N)}$ have taken into account the conditional messages related to both $s$ and $q$. In addition, it encodes only the $N$-hop neighbor entities of the subject entity. We assign a score of 0 to other entities.

### 4.4 Pre-training Objective

Given a set of source KGs $\mathcal{C} = \{\mathcal{K}_0, \ldots, \mathcal{K}_n\}$, where $\mathcal{K}_i = (\mathcal{E}_i, \mathcal{R}_i, \mathcal{T}_i)$, we pre-train the model using the multi-class log-loss [51]:

$$\sum_{(\mathcal{E}_i, \mathcal{R}_i, \mathcal{T}_i) \in \mathcal{C}} \sum_{(s,q,o) \in \mathcal{T}_i} \bigg( -f(s, q, o) + \log\Big(\sum_{e \in \mathcal{E}_i} \exp\big(f(s, q, e)\big)\Big)\bigg), \tag{9}$$

where $f$ is the score function mentioned above. Minimizing Equation (9) enlarges scores of positive facts while reducing scores of all negatives that replace the correct object entity with another entity from the KG. We describe our reasoning process in Algorithm 1 of Appendix B.

## 5 Experiments

### 5.1 Settings

**Datasets and implementations.** We conduct experiments on 43 datasets of various schemata and sizes to evaluate our model. The datasets fall into three groups: (i) 14 inductive datasets, including 12 datasets in GraIL [1] and 2 datasets in ILPC 2022 [52], (ii) 13 fully-inductive datasets in [5], and (iii) 16 transductive datasets, including FB15k-237 [53], WN18RR [12], NELL-995, [54], YAGO3-10 [55], 3 datasets in CoDEx [56], 5 datasets in [57], AristoV4 [58], DBpedia100k [59], ConceptNet100k [60], and Hetionet [61]. The statistics of datasets are in Appendix F. We pre-train our model on three datasets, i.e., FB V1 [1] with 180 relations, NELL V1 [1] with 14 relations, and CoDEx-s [56] with 42 relations, to capture various relational structures in KGs and prompt graphs. The implementation details are in Appendix C. We assess the impact of pre-training KGs in Appendix E.1.

**Baselines.** We compare KG-ICL with two categories of baseline models: (i) Supervised state-of-the-art (abbr. supervised SOTA), which refers to the models achieving the best MRR result on specific target datasets. We list the supervised SOTA model on each dataset in Appendix F. (ii) Pre-training model. ULTRA [6] is a KG pre-training model, consisting of pre-training and finetuning versions. To investigate the ability of finetuning on target datasets to yield improvement of the proposed model, we also introduce two versions of our model: "KG-ICL pre-train" and "KG-ICL finetune". After pre-training, the finetuning model undergoes finetuning for 5 epochs on the target dataset using the same configuration as the pre-training. The main focus of this work is on in-context learning without

the need for finetuning. Therefore, we report the results of both versions of our model in Section 5.2 and Appendix G, and use the pre-training version in further analyses.

**Evaluation protocol.** For each sample $(s, r, o)$ in the test set, we generate two query facts $(s, r, ?)$ and $(o, r^-, ?)$, where $r^-$ is the inverse relation of $r$. As mentioned in Section 3, we add inverse relations and facts before conducting reasoning. The pre-training model considers all entities in the entity set as candidates, scoring and ranking them for each query fact. Following the convention [62, 63, 64], we employ two standard evaluation metrics: mean reciprocal rank (MRR) and Hits@10 (abbr. H@10). Higher scores of both metrics indicate superior performance. We follow the widely-used filtered setting [10], i.e., wherein all known true entities are removed from the candidate set, except for the target entity. Due to the abundance of datasets, we categorize them and report the scores in terms of the groups, such as the inductive dataset group. This involves calculating scores for each dataset individually and computing the average of all scores within each group.

Table 1: KG reasoning results in various settings.

| Models | Inductive (14 KGs) | | Fully-inductive (13 KGs) | | Transductive (16 KGs) | | Average (43 KGs) | |
|---|---|---|---|---|---|---|---|---|
| | MRR | H@10 | MRR | H@10 | MRR | H@10 | MRR | H@10 |
| Supervised SOTA | 0.466 | 0.607 | 0.210 | 0.347 | 0.365 | 0.511 | 0.351 | 0.493 |
| ULTRA pre-train | 0.513 | 0.664 | 0.352 | 0.536 | 0.329 | 0.479 | 0.396 | 0.557 |
| ULTRA finetune | 0.528 | 0.684 | 0.350 | 0.542 | 0.384 | 0.548 | 0.421 | 0.590 |
| KG-ICL pre-train | 0.554 | 0.707 | 0.439 | 0.635 | 0.346 | 0.493 | 0.442 | 0.606 |
| KG-ICL finetune | **0.582** | **0.727** | **0.449** | **0.647** | **0.397** | **0.554** | **0.473** | **0.638** |

## 5.2 Main Results

We divide the datasets into three groups according to their reasoning settings, i.e., inductive, fully-inductive and transductive, and report the average results for each group as well as the overall average results in Table 1. ULTRA employs three different source KGs distinct from ours. For ease of presentation, we incorporate the source KGs into their respective groups rather than listing them separately. We can observe that our "KG-ICL pre-train" outperforms both versions of ULTRA on inductive and fully-inductive datasets, with further enhancements achieved by our "KG-ICL finetune", resulting in the best performance across all groups. We report detailed results of each dataset and more analyses in Figure 2 and Appendix G.

**Inductive datasets.** The inductive setting aims to complete facts involving unseen entities. In each inductive dataset, at least two graphs are included: one for training and the other for evaluation. The evaluation graph incorporates new entities not seen in the training graph. The MRR results are depicted in Figure 2(a). We observe that our "KG-ICL pre-train" outperforms supervised SOTA models on 10 datasets and surpasses the "ULTRA pre-train" model on 11 datasets. The "KG-ICL finetune" achieves further improvements in scores compared to the pre-trained version, achieving the best results on 13 datasets, and yielding the best average results.

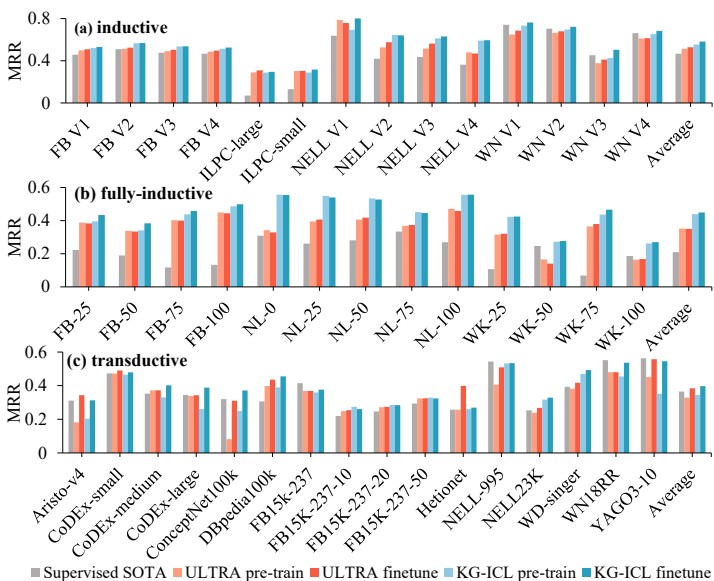

Figure 2: MRR results on various KGs.

**Fully-inductive datasets.** In fully-inductive datasets, the evaluation graphs not only include new entities unseen during training but also introduce new relations. The MRR results are shown in Figure 2(b). This setting poses a significant challenge with the introduction of unseen relations, leading to relatively lower scores for supervised SOTA models. However, both versions of KG-ICL, aided by prompt graphs, demonstrate the ability to extract valuable information and adaptively perform reasoning for unseen query relations. Consequently, they consistently outperform supervised SOTA models across all 13 datasets and exhibit a notable improvement over the previous pre-training model, ULTRA, on all datasets.

**Transductive datasets.** In the transductive setting, where entities and relations in the test set are encountered during training, the MRR results are presented in Figure 2(c). It is evident that, in comparison to the first two settings, the advantage of the proposed model over the supervised SOTA model is somewhat attenuated. The reason is that the supervised signals on transductive datasets directly target entities and relations in the test set, allowing supervised models to effectively learn representations and achieve high performance. Nonetheless, "KG-ICL pre-train" maintains its superiority over the supervised SOTA models on 7 datasets. "KG-ICL finetune" achieves the best average MRR score. We report detailed results of each dataset and more analyses in Appendix G.

### 5.3 Further Analyses

In this section, we conduct experiments to devise the impact of each module. In the appendix, we include more experimental analyses about the pre-training datasets (Appendix E.1), complexity analyses of preprocessing (Appendix E.2), and the variant incorporating other message passing layer (Appendix E.3).

Table 2: Ablation study results in various settings.

| Models | Inductive (14 KGs) | | Fully-inductive (13 KGs) | | Transductive (16 KGs) | | Average (43 KGs) | |
|---|---|---|---|---|---|---|---|---|
| | MRR | H@10 | MRR | H@10 | MRR | H@10 | MRR | H@10 |
| Intact model | 0.554 | 0.707 | 0.439 | 0.635 | 0.346 | 0.493 | 0.442 | 0.606 |
| w/o prompt graph | 0.219 | 0.420 | 0.105 | 0.228 | 0.076 | 0.143 | 0.132 | 0.259 |
| w/o unified tokenizer | 0.511 | 0.660 | 0.419 | 0.617 | 0.296 | 0.453 | 0.403 | 0.570 |
| w/ GraIL's labeling | 0.531 | 0.704 | 0.434 | 0.634 | 0.343 | 0.492 | 0.431 | 0.604 |

**Ablation study.** We hereby conduct an ablation study to evaluate the impact of each module. Specifically, we construct three variants by removing certain modules: "w/o prompt graph", "w/o unified tokenizer", and "w/ GraIL's labeling". "w/o prompt graph" removes the prompt graph generation and encoding module. Its prompt representations are initialized with the Xavier normal initialization. "w/o unified tokenizer" eliminates the unified tokenizer and initializes the input representations of entities and relations in prompt graphs with the Xavier normal initialization. "w/ GraIL's labeling" replaces our token representation with GraIL's one-hot labeling [1]. The results are presented in Table 2. We observe a significant performance decline in the "w/o prompt graph" variant compared to the intact model, highlighting the necessity of the prompt graph as a knowledge transfer bridge. The "w/o unified tokenizer" variant also exhibits a performance drop, indicating the importance of the unified tokenizer for in-context learning. The "w/ GraIL's labeling" can also achieve promising results, although it still falls behind our intact model, which shows the generalization ability of our model and the effectiveness of the token representation.

**Example efficiency.** The efficiency of utilizing examples is crucial for in-context learning. To determine the optimal number of example prompt graphs needed to support in-context reasoning, we conduct experiments under the settings of 1-shot, 3-shot, 5-shot, 10-shot, and 20-shot. The results are illustrated in Figure 3. Overall, the results remain consistent with a slight fluctuation across the range from 1-shot to 20-shot, which shows that the proposed model is robust to the changes in the num-

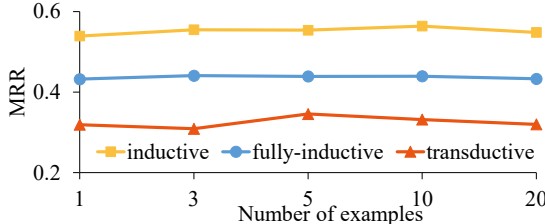

Figure 3: MRR with different numbers of examples.

ber of prompt graphs. The reason for the slight performance fluctuation is that more examples may also introduce more noise. Besides, multiple examples tend to share popular reasoning patterns, so only one or three prompt graphs can still suffice. For overall performance, we choose $M = 5$ in the main experiment. These results suggest that KG-ICL can unleash universal reasoning capabilities with only a few examples, showcasing high efficiency in example utilization.

**Prompt graph variants.** The core of a prompt graph lies in highlighting essential information for reasoning. In this paper, we propose a prompt graph generation process that combines paths and neighbors of the subject and object entities. To further explore the critical components for reasoning, we introduce several variants, with the proposed model referred to as "neighbor & 3-hop path". We present four variants by altering the entity sampling method: the "neighbor" variant, considering only neighbors of the subject and object entities, and the "$x$-hop path" variant, considering $x$-hop paths between the entity and object entities, where $x \in \{1, 2, 3\}$. The results in Table 3 demonstrate the impact of the prompt graph on reasoning. We observe that both paths and neighbors of the subject and object entities are crucial for reasoning. The optimal performance is achieved when combining both components. The variants considering only paths within one or two hops exhibit poor performance, indicating insufficient support for effective reasoning.

Table 3: MRR results on diverse prompt graphs.

| Models | Inductive (14 KGs) | | Fully-inductive (13 KGs) | | Transductive (16 KGs) | | Average (43 KGs) | |
|---|---|---|---|---|---|---|---|---|
| | MRR | H@10 | MRR | H@10 | MRR | H@10 | MRR | H@10 |
| Neighbor & 3-hop path | 0.554 | 0.707 | 0.439 | 0.635 | 0.346 | 0.493 | 0.442 | 0.606 |
| Neighbor | 0.552 | 0.702 | 0.429 | 0.628 | 0.311 | 0.459 | 0.425 | 0.590 |
| 1-hop path | 0.208 | 0.449 | 0.145 | 0.314 | 0.112 | 0.216 | 0.153 | 0.322 |
| 2-hop path | 0.256 | 0.419 | 0.137 | 0.285 | 0.125 | 0.235 | 0.171 | 0.310 |
| 3-hop path | 0.544 | 0.697 | 0.409 | 0.601 | 0.294 | 0.464 | 0.410 | 0.582 |

**Robustness to low-resource relations.** We conduct experiments to assess the robustness of the proposed model to low-resource relations with limited training samples. Specifically, we choose the supervised model RED-GNN [3] as a baseline and conduct experiments on 12 widely used inductive datasets [1] and 3 transductive datasets (FB15k-237, WN18RR, and NELL-995). We organize relations within each dataset group into six subgroups based on the number of training samples. Subsequently, we compute the MRR score for each relation and calculate the average score within each subgroup. The results, as illustrated in Figure 4, reveal a gradual decline in the performance of RED-GNN, as the number of training samples decreases. In contrast, our model exhibits robustness across a spectrum of relations. The results suggest that our model maintains effective performance even under resource constraints. This can be attributed to our model of avoiding the representation of each relation independently with specific embeddings. We employ a universal prompt graph and a unified tokenizer for the relation representation, fostering cross-relation knowledge transfer and achieving superior robustness.

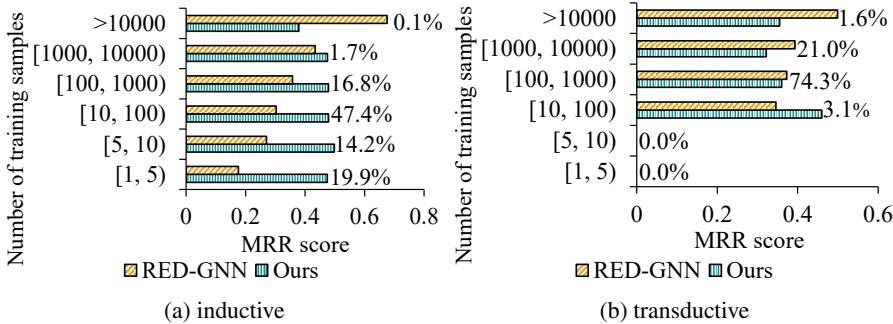

(a) inductive                    (b) transductive

Figure 4: Average MRR results of relation subgroups. Relations in the inductive and transductive dataset groups are divided into 6 subgroups based on the number of training samples, and the results represent the average scores for the relations within their respective subgroups. The percentage on the right side of each data bar indicates the proportion of relations in that subgroup to the total number of relations in their respective groups.

**Case study.** We conduct a case study to investigate the reasons behind the proposed model's generalizability across different KGs. Specifically, we select two similar and easily interpretable query relations, "teamSport" and "film/language" from NELL-995 and FB15k-237, respectively. We extract several relation paths from their prompt graphs, forming two similar subgraphs. Subsequently, we execute the model and save the prompt representations for both query relations. Finally, we compute the cosine similarities between relations in the two prompt graphs and visualize the heatmap in Figure 5. We observe that the values along the diagonal of the heatmap are notably high, indicating that different relations with similar roles in the reasoning of the two query relations have correspondingly similar model encodings. This suggests that the prompt representations effectively capture the roles of various relations in reasoning, thereby improving transferability across different KGs.

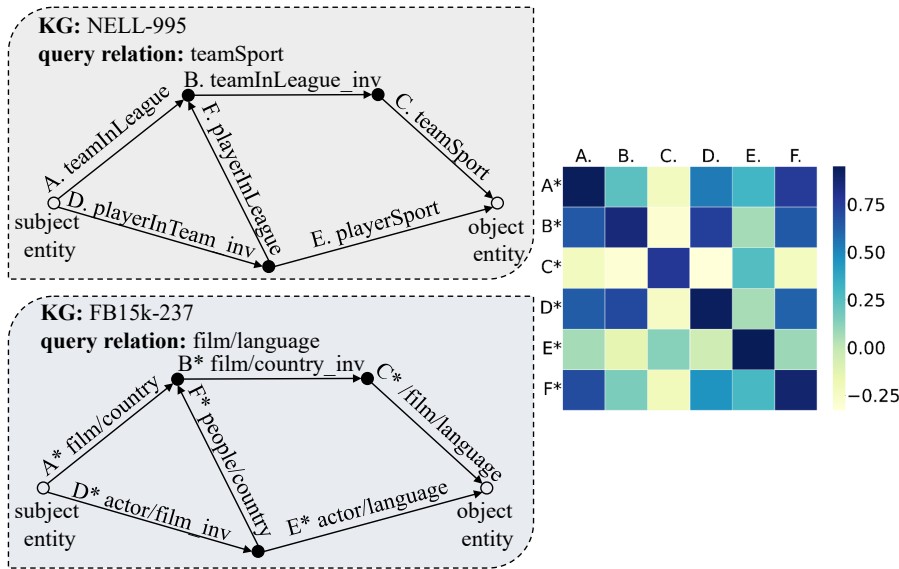

Figure 5: Case study on prompt graphs. The left side shows some relation paths extracted from two prompt graphs of NELL-995 and FB15k-237. The right side depicts a heatmap where cosine similarities between relations in two prompt graphs are pairwise computed.

## 6  Conclusions

This paper introduces a KG foundation model with in-context learning to improve the effectiveness and transferability of KG reasoning. Specifically, we introduce a prompt graph and a unified tokenizer as the bridge to knowledge transfer between different KGs. Following that, we propose a prompt graph generation module, a prompt encoding module, and a KG reasoning module to achieve in-context learning. We evaluate the in-context reasoning ability on 43 different KGs in both transductive and inductive settings. Extensive experimental results validate our model's universal reasoning ability across diverse KGs. In future work, we plan to explore the application of in-context reasoning in more challenging scenarios, such as personal KGs that are dynamic and diverse. This is motivated by the demonstrated robustness of our KG-ICL in Section 5.3. Additionally, investigating how to extend in-context reasoning to more knowledge-driven applications, e.g., recommender systems and question answering, is another promising avenue for future research.

## Acknowledgments

We thank the anonymous reviewers for their valuable comments. This work was funded by National Natural Science Foundation of China (Nos. 62272219 and 62406136), Postdoctoral Fellowship Program of CPSF (No. GZC20240685), and CCF-Tencent Rhino-Bird Open Research Fund.

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

## A   Message Passing Architectures

### A.1   Message Passing Neural Network for Prompt Encoding

Based on the framework mentioned in Section 4.2, we present two types of aggregation: entity-centric and relation-centric aggregations. In each layer, we first update the entity representations and then update the relation representations. Specifically, given a central entity $e$ and the query relation $q$, we update the representation of $e$ using following entity-centric aggregation function:

$$\mathbf{e}^{(l+1)} = \mathrm{ReLU}\Big( \text{Max-pooling}\,\big\{\mathbf{m}_{s,r,q}\,|\,(s,r,e) \in \mathcal{N}_e\big\}\Big), \tag{10}$$

$$\mathbf{m}_{s,r,q} = \alpha_{r;q}\mathbf{W}_{\text{E-msg}}^{(l)}\Big(\mathbf{s}^{(l)}\,||\,\mathbf{r}^{(l)}\,||\,\mathbf{q}^{(l)}\Big), \tag{11}$$

$$\alpha_{r;q} = \sigma\Big(\mathbf{W}_{\text{E-attn}}^{(l)}\big(\mathbf{r}^{(l)}\,||\,\mathbf{q}^{(l)}\big)\Big), \tag{12}$$

where $\mathcal{N}_e \subseteq \mathcal{T}_{\text{pmt}}$ is the set of fact containing $e$. $\mathbf{W}_{\text{E-msg}}^{(l)} \in \mathbb{R}^{d\times 3d}$ and $\mathbf{W}_{\text{E-attn}}^{(l)} \in \mathbb{R}^{1\times 2d}$ are two learnable parameter matrices. $\mathbf{s}^{(l)}, \mathbf{r}^{(l)}, \mathbf{q}^{(l)}$ are the representations of $s, r, q$ in the $l$-th layer, separately. $(\cdot||\cdot)$ denotes the concatenate operation. $\sigma(\cdot)$ denotes the Sigmoid activation function.

We also adopt a query-aware attention mechanism for the relation-centric aggregation:

$$\mathbf{r}^{(l+1)} = \mathrm{ReLU}\Big( \text{Max-pooling}\,\big\{\mathbf{m}_{s,o,q}\,|\,(s,r,o) \in \mathcal{N}_r\big\}\Big) + \mathbf{r}^{(l)}, \tag{13}$$

$$\mathbf{m}_{s,o,q} = \alpha_{r;q}\mathbf{W}_{\text{R-msg}}^{(l)}\Big(\mathbf{s}^{(l+1)}\,||\,\mathbf{o}^{(l+1)}\,||\,\mathbf{q}^{(l)}\Big), \tag{14}$$

$$\alpha_{r;q} = \sigma\Big(\mathbf{W}_{\text{R-attn}}^{(l)}\big(\mathbf{r}^{(l)}\,||\,\mathbf{q}^{(l)}\big)\Big), \tag{15}$$

where $\mathcal{N}_r \subseteq \mathcal{T}_{\text{pmt}}$ is the set of fact containing $r$. $\mathbf{W}_{\text{R-msg}}^{(l)} \in \mathbb{R}^{d\times 3d}$ and $\mathbf{W}_{\text{R-attn}}^{(l)} \in \mathbb{R}^{1\times 2d}$ are two learnable parameter matrices, and $\mathbf{o}^{(l+1)} \in \mathbf{H}_{\text{E}}^{(l+1)}$ is the representation of $o$.

We also incorporate residual connection [65] and layer normalization [66] to enhance learning.

### A.2   Message Passing Neural Network for KG Encoding

Based on the framework mentioned in Section 4.3, we present a message passing neural network for KG encoding. Specifically, given a central entity $e$ and the query relation $q$, we update the representation of $e$ using following aggregation function:

$$\begin{aligned}
\mathbf{e}^{(l+1)} &= \mathrm{ReLU}\Big( \text{Mean-pooling}\{\mathbf{m}_{s,r,q}\,|\,(s,r,o) \in \mathcal{N}_e\}\Big), \\
\mathbf{m}_{s,r,q} &= \alpha_{s;r;q}\mathbf{W}_{\text{msg}}^{(l)}\Big(\mathbf{s}^{(l)} + \mathbf{r}^{(l+1)}\Big), \\
\alpha_{s;r;q} &= \sigma\Big(\mathbf{W}_{\text{attn}}^{(l)}\Big(\mathbf{W}_{\text{s}}^{(l)}\mathbf{s}^{(l)}\big) + \mathbf{W}_{\text{r}}^{(l)}\mathbf{r}^{(l+1)} + \mathbf{W}_{\text{q}}^{(l)}\mathbf{q}^{(l+1)}\Big)\Big),
\end{aligned} \tag{16}$$

where $\mathcal{N}_e \subseteq \mathcal{T}$ is the set of fact containing $e$, $\mathbf{W}_{\text{msg}}^{(l)}, \mathbf{W}_{\text{attn}}^{l} \in \mathbb{R}^{d\times d}$ and $\mathbf{W}_{\text{s}}^{(l)}, \mathbf{W}_{\text{r}}^{(l)}, \mathbf{W}_{\text{q}}^{(l)} \in \mathbb{R}^{1\times d}$ are learnable parameter matrices, $\mathbf{s}^{(l)}, \mathbf{r}^{(l+1)}, \mathbf{q}^{(l+1)}$ are the representations of $s, r, q$, separately.

## B   In-Context Reasoning Pipeline

We integrate the modules in Section 4 together and present a pipeline of in-context reasoning in Algorithm 1. Given an input query fact and its corresponding KG, we perform reasoning by scoring all candidate entities. In Lines 1–2, we first generate a few prompt graphs as context of the query relation and map entities and relations within them to predefined tokens. In Lines 3–9, we encode the prompt graphs to obtain prompt representations. In practice, we parallel encode these prompt graphs to ensure efficiency. In Line 10, we initialize the representations of KG entities and relations based on the prompts. In Lines 11–13, a multi-layer message passing neural network is employed for KG encoding. In Line 14, we assign a score for each candidate entity based on the output entity representations. For inference, these scores are used for entity ranking and metric calculation. For pre-training, these scores are utilized in Equation (9) to obtain the loss and update model parameters.

---

**Algorithm 1:** In-context reasoning

---

**Input:** A query $(s, q, ?)$ and the KG $\mathcal{K} = (\mathcal{E}, \mathcal{R}, \mathcal{T})$ it within.
**Output:** Scores of all candidate entities in $\mathcal{E}$.

```
/* Stage 1:  Prompt graph generation                                    */
```
1  Generate $M$ prompt graphs $\{\mathcal{P}_{c_1}, \ldots, \mathcal{P}_{c_M}\}$ for the query relation $q$;
2  Map the entities and relations in prompt graphs to predefined tokens using unified tokenizer;
```
/* Stage 2:  Prompt encoding                                            */
```
3  **for** $\mathcal{P}_{c_i} \leftarrow 1$ **to** $M$ **do**
4       Initialize entity and relation representations $\mathbf{H}_{\mathrm{E}}^{(0)}$ and $\mathbf{H}_{\mathrm{R}}^{(0)}$ using token representations;
5       **for** $l \leftarrow 1$ **to** $L$ **do**
6           Update entity representations $\mathbf{H}_{\mathrm{E}}^{(l)}$ using Equation (3);
7           Update relation representations $\mathbf{H}_{\mathrm{R}}^{(l)}$ using Equation (4);
8           Readout the relation representations $\mathbf{H}_{\mathcal{P}_{c_i}}$ using Equation (5);
9       Obtain prompt representation matrix $\overline{\mathbf{H}}_{\mathrm{pmt}}$ using Equation (6);
```
/* Stage 3:  KG encoding and reasoning                                  */
```
10 Initialize entity and relation representations $\mathcal{V}_{\mathrm{E}}^{(0)}$ and $\mathcal{V}_{\mathrm{R}}^{(0)}$ based on $\overline{\mathbf{H}}_{\mathrm{pmt}}$;
11 **for** $l \leftarrow 1$ **to** $N$ **do**
12      Update relation representations $\mathbf{V}_{\mathrm{R}}^{(l)}$ using Equation (7);
13      Update entity representations $\mathbf{V}_{\mathrm{E}}^{(l)}$ using Equation (8);
14 Score entities in $\mathcal{E}$ based on entity representations $\mathbf{V}_{E}^{(N)}$ using reasoning module in Section 4.3;

---

## C  Implementation Details

Under the framework in Section 4, we implement an in-context reasoning model KG-ICL, which employs a 5-shot 3-hop prompt graph as context, along with 3 stacked layers for prompt graph encoding, and 6 stacked layers for KG encoding and reasoning, i.e., $M = 5$, $k = 3$, $L = 3$ and $N = 6$. The dimension $d$ of the hidden layers is set to 32. Following the standard in-context learning process [46], we first pre-train a model on source datasets and then freeze the model parameters for evaluation. We pre-train our model on three source datasets, i.e., FB V1 [1] with 180 relations, NELL V1 [1] with only 14 relations, and CoDEx-small [56] with 42 relations. We use Adam optimizer and set the learning rate to 0.001 and the patience of early stopping to 5. The pre-training process is conducted on a workstation with two Intel Xeon Gold CPUs, four NVIDIA RTX A6000 GPUs, and Ubuntu 18.04 LTS. The pre-training model maintains a modest size with only 89k parameters, and the pre-training process converges in less than six hours.

## D  Related Work

### D.1  Knowledge Graph Reasoning

*Diverse KG reasoning settings.* KG reasoning primarily involves three settings: transductive, inductive, and fully-inductive. Early studies [10, 11, 12, 13, 14] focus mainly on the transductive setting, assuming that KGs are static. They learn an embedding for each specific entity, making it challenging to handle the addition of new entities. Real-world KGs are dynamic, inspiring the development of inductive models [1, 2, 3, 4, 15, 16, 17, 18, 19, 20, 21, 23] that allows for unseen entities. These models base their reasoning on relation patterns rather than entity embeddings. In the fully-inductive setting [5, 24, 25, 26], unseen entities and relations can both emerge in the query facts. While this setting is closer to pre-training, it remains limited to the same KG. The distinction among these settings arises from the fact that text data can be naturally split into unified tokens, while the entity and relation sets across KGs are not shared. In this paper, we propose a prompt graph and a unified tokenizer to support in-context learning, breaking down the barriers imposed by these settings and achieving universal reasoning capabilities.

*Entity alignment and pre-training.* Extensive research efforts have been concentrated on establishing a unified entity vocabulary to support pre-training through the recognition of identical entities in

different KGs, a task commonly known as entity alignment [18, 67, 68, 69, 70, 71, 72]. Based on these aligned entities, some KG pre-training models [34, 73, 74, 75, 76, 77] have been proposed to map the entities in diverse KGs into a unified semantic space. Nevertheless, this paradigm heavily depends on expensive pre-labeled alignment, which is not always sufficient in the real world. More critically, it relies on similar schemata [34], lacking robust generality across KGs that span diverse domains. ULTRA [6] introduces a foundation model following the paradigm of "pre-training then finetuning", which is an alignment-free reasoning model. Stand on the shoulders of previous work, we aim to avoid dataset-specific finetuning and propose a model that can achieve universal reasoning capabilities with just a few examples as contextual prompts. In Section 5, we conduct comparative experiments with ULTRA, and the results demonstrate the effectiveness of our in-context learning. Additionally, KGTransformer [78] introduces a prompt-based KG pre-training model to support a variety of downstream tasks. Their objective is not KG reasoning but rather the transfer of knowledge from KGs to enhance downstream tasks such as image classification.

### D.2 Prompt-based In-Context Learning

Our work is also related to prompt learning and in-context learning. Here, our primary focus is on KG reasoning, which shares similar challenges with graph learning. Drawing inspiration from the success of early pre-training models in NLP [27] and computer vision [28], some graph pre-training models [29, 30, 31, 32, 33] have been proposed. These models follow the paradigm of "pre-train and finetune", where a model is initially pre-trained and then finetuned for the target task. However, this paradigm has limitations in terms of generalization and may sometimes lead to negative knowledge transfer. Consequently, recent researches [8, 35, 36, 37, 38, 39, 40, 41, 42, 43, 44, 45] have shifted focus towards the "pre-train, prompt, and finetune" paradigm. This paradigm leverages task prompts to enhance the knowledge transfer and generalization capabilities of pre-trained models, achieving significant progress. KG reasoning involves making inferences based on multi-relational data. Therefore, these pre-trained models are not easily applicable to KG reasoning tasks. Inspired by the success of recent black-box large language models like GPT [7], the in-context learning paradigm aims to avoid finetuning on the target dataset. Instead, it imparts general capabilities to pre-trained models with just a few examples. PRODIGY [46] introduces an in-context learning-based graph pre-training model to handle various classification tasks. While it can perform relation classification tasks, it is not suitable for KG reasoning with a massive number of candidate entities.

## E  Further Analyses

### E.1  Impact of Pre-training Mixture

The effectiveness of in-context learning is inherently tied to the quality and diversity of the source datasets used for pre-training. Here, we analyze the impact of the bias of source KGs by introducing six different combinations of source KGs. The results are reported in Table 4. We observe that (i) more source KGs help reduce the influence of biases in individual datasets, and (ii) these three source KGs are of good quality, as even using just one for pre-training yields decent performance. Besides, we find that our pre-training does not require a large scale of KG facts. The variety of relational structures is more important for our pre-training. Thus, in practice, we can choose several KGs with different schemata or from different domains for pre-training.

Table 4: Performance w.r.t. different pre-training KGs.

| Source Datasets | | | Average Results | |
| --- | --- | --- | --- | --- |
| FB V1 | NELL V1 | CoDEx-small | MRR | H@10 |
| ✓ | | | 0.424 | 0.586 |
| | ✓ | | 0.392 | 0.565 |
| | | ✓ | 0.389 | 0.561 |
| ✓ | ✓ | | 0.425 | 0.592 |
| ✓ | | ✓ | 0.436 | 0.606 |
| | ✓ | ✓ | 0.423 | 0.595 |
| ✓ | ✓ | ✓ | 0.442 | 0.606 |

We also conducted experiments using the same pre-training mixture as ULTRA [6], specifically WN18RR, FB15k-237, and CoDEx-medium, to pre-train KG-ICL. The results are shown in Table 5. We observe that KG-ICL outperforms ULTRA in this setting. Pre-training with these datasets causes a slight decrease in the inductive performance and a slight improvement in the transductive results, but neither change is significant. We use three smaller datasets to pre-train KG-ICL, as smaller datasets do not significantly affect model performance and can expedite the pre-training process.

Table 5: Performance w.r.t the same pre-training mixture as ULTRA [6].

| Models | Inductive (14 KGs) | | Fully-inductive (13 KGs) | | Transductive (16 KGs) | | Average (43 KGs) | |
| --- | --- | --- | --- | --- | --- | --- | --- | --- |
| | MRR | H@10 | MRR | H@10 | MRR | H@10 | MRR | H@10 |
| ULTRA (FB15k-237 WN18RR CoDEx-medium) | 0.513 | 0.664 | 0.352 | 0.536 | 0.329 | 0.479 | 0.396 | 0.557 |
| KG-ICL (FB15k-237 WN18RR CoDEx-medium) | 0.547 | 0.700 | 0.431 | 0.629 | 0.357 | 0.506 | 0.441 | 0.606 |
| KG-ICL (FB V1 NELL V1 CoDEx-small) | 0.554 | 0.707 | 0.439 | 0.635 | 0.346 | 0.493 | 0.442 | 0.606 |

Following [6], we conduct experiments on growing pre-training mixtures, sequentially adding pre-training datasets in the same order as in [6], i.e., FB15k-237, WN18RR, CoDEx-medium, NELL-995, YAGO3-10, ConceptNet100K, DBpedia100K, and AristoV4. The results are shown in Figure 6. We observe that the performance improves with the number of pre-training datasets. Unlike ULTRA, KG-ICL even performs well with pre-training on a single KG. This improvement is due to two key factors: first, we generate a diverse set of prompt graphs for different relations within the same KG, which increases sample diversity. Second, our targeted prompt engineering reduces learning complexity and facilitates better generalization.

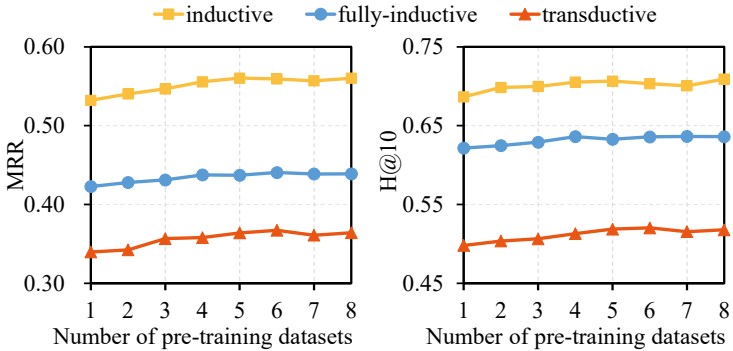

Figure 6: MRR and H@10 results with increasing number of pre-training datasets.

### E.2 Complexity Analyses of Prompt Graph Preprocessing

The proposed model relies on generating and processing prompt graphs. Here, we analyze the computational complexity of the pre-processing efficiency. The generation processing of a prompt graph includes two steps: (i) Subgraph extraction. We take the intersection of the $k$-hop neighbors of the subject entity $u$ and the object entity $v$ to obtain the set of nodes ($O(|\mathcal{T}| + |\mathcal{E}|)$), where $\mathcal{T}$ and $\mathcal{E}$ are the set of facts and entities in the KG, respectively. Then, we retrieve the facts between these entities ($O(|\mathcal{T}|)$). (ii) Labeling. We perform twice single-source shortest path length calculations (for $u$ and $v$) on the prompt subgraph ($O(|\mathcal{T}_{\mathrm{pmt}}| + |\mathcal{E}_{\mathrm{pmt}}|) \times \log|\mathcal{E}_{\mathrm{pmt}}|$) to get token labels. In summary, the overall computational complexity is $O(|\mathcal{T}| + |\mathcal{E}| + (|\mathcal{T}_{\mathrm{pmt}}| + |\mathcal{E}_{\mathrm{pmt}}|) \times \log|\mathcal{E}_{\mathrm{pmt}}|)$. Note that the size of the prompt graph is usually much smaller than the entire KG. In the $M$-shot (e.g., 5-shot) in-context learning setting, we only need to extract $M \times |\mathcal{R}|$ prompt graphs, where $\mathcal{R}$ denotes the set of relations in KG. We report the preprocessing times under the 5-shot setting in Table 6. We can observe that preprocessing all 43 datasets (including some large KGs) requires 1597 seconds, averaging 37.1 seconds per dataset. Among them, AristoV4 [58], with the most relations (with 1605 relations, 44949 entities, and 242567 facts), has the longest preprocessing time, at 995 seconds, averaging 0.62 seconds per relation. Hetionet [61], with the most facts (with 24 relations, 45158 entities, and 2025177 facts), has a preprocessing time of 120 seconds, averaging 5 seconds per relation. YAGO3-10 [55], with the most entities (with 34 relations, 123182 entities, and 1079040

facts), has a preprocessing time of 50 seconds, averaging 1.47 seconds per relation. In summary, the preprocessing method is scalable because we extract only a small number of examples for each relation. Evaluations on large-scale datasets containing millions of facts also confirm its scalability.

Table 6: The total and average preprocessing time.

| Datasets | Time (s) | |
|---|---|---|
| | Total | Average |
| Inductive (14 KGs) | 42.0 | 3.0 |
| Fully-inductive (13 KGs) | 96.0 | 7.4 |
| Transductive (16 KGs) | 1459.0 | 91.2 |
| All (43 KGs) | 1597.0 | 37.1 |

### E.3 Incorporating Other Message Passing Layer

The proposed model can also be incorporated with other message passing neural networks that can aggregate messages conditioned with specific queries. Here, we implement a variant, KG-ICL (NBFNet), by incorporating the message passing of NBFNet [4], which is used by ULTRA [6]. Note that NBFNet only outputs entity representations but not updates or outputs relation representations, which is also one reason we did not adopt NBFNet initially. ULTRA treats relations as nodes to obtain relation representations. Therefore, we incorporate Equation (4) into NBFNet (default configuration) to support relation encoding. The results are shown in Table 7. We can observe that KG-ICL (NBFNet) also achieved promising results, slightly below KG-ICL. This demonstrates the potential of combining KG-ICL with more passing message neural networks. Moreover, the structure of this variant is similar to ULTRA, but the input is prompt graphs rather than relation graphs, which indicates the superiority of our prompt graph to that of ULTRA's relation graph in relation modeling.

Table 7: The results of the variant incorporating with NBFNet.

| Models | Inductive (14 KGs) | | Fully-inductive (13 KGs) | | Transductive (16 KGs) | | Average (43 KGs) | |
|---|---|---|---|---|---|---|---|---|
| | MRR | H@10 | MRR | H@10 | MRR | H@10 | MRR | H@10 |
| KG-ICL | 0.554 | 0.707 | 0.439 | 0.635 | 0.346 | 0.493 | 0.442 | 0.606 |
| KG-ICL (NBFNet) | 0.545 | 0.703 | 0.423 | 0.622 | 0.298 | 0.438 | 0.416 | 0.580 |
| ULTRA | 0.513 | 0.664 | 0.352 | 0.536 | 0.329 | 0.479 | 0.396 | 0.557 |

## F   Dataset Statistics

We conduct extensive evaluations on 43 datasets. We categorize the datasets into three types: inductive datasets, fully-inductive datasets, and transductive datasets. The statistical data of these datasets and their state-of-the-art models are reported in Tables 8, 9, and 10, respectively.

## G   Detailed Results

To validate the effectiveness of our in-context reasoning model, we compare KG-ICL with the supervised SOTA models and ULTRA's pre-training and finetuning versions on 43 datasets. The detailed results for each dataset are presented in Table 11. We observe that (i) KG-ICL outperforms the competitors on most datasets, demonstrating the universal reasoning capability of our in-context model. (ii) "KG-ICL pre-train" outperforms "ULTRA pre-train" on 11 inductive datasets, all 13 fully-inductive datasets, and 9 transductive datasets, which demonstrates the superiority of our in-context KG reasoning foundation model. In addition, "KG-ICL finetune" also outperforms "ULTRA finetune" on most datasets. (iii) InGram [5] transfers knowledge to new query relations through a relation graph, while we employ the prompt graph as a bridge for knowledge transfer. Our KG-ICL outperforms InGram on all 13 fully-inductive datasets, indicating that our prompt graphs can better highlight important clues for specific query relations than relation graphs. (iv) On transductive datasets, KG-ICL's performance improvement compared to supervised baseline models is less than that in the previous two settings. There are two reasons for this: first, the supervised signals on transductive datasets directly target entities and relations in the test set, allowing supervised models to

Table 8: Statistics of inductive datasets.

| Datasets | #Rel. | Training graphs | | Validation graphs | | | Test graphs | | | SOTA |
|---|---|---|---|---|---|---|---|---|---|---|
| | | #Ent. | #Facts | #Ent. | #Facts | #Valid. | #Ent. | #Facts | #Test | |
| FB V1 [1] | 180 | 1594 | 4245 | 1594 | 4245 | 489 | 1093 | 1993 | 411 | A*Net [21] |
| FB V2 [1] | 215 | 3668 | 9799 | 3668 | 9799 | 1166 | 2501 | 4406 | 947 | NBFNet [4] |
| FB V3 [1] | 215 | 3668 | 17986 | 3668 | 17986 | 2194 | 2501 | 7406 | 1731 | NBFNet [4] |
| FB V4 [1] | 219 | 4707 | 27203 | 4707 | 27203 | 3352 | 3051 | 11714 | 2840 | A*Net [21] |
| WN V1 [1] | 9 | 2746 | 5410 | 2746 | 5410 | 630 | 922 | 1618 | 373 | NBFNet [4] |
| WN V2 [1] | 10 | 6054 | 15606 | 6054 | 15606 | 1838 | 2757 | 4011 | 852 | NBFNet [4] |
| WN V3 [1] | 11 | 12078 | 25901 | 12078 | 25901 | 3097 | 5084 | 6327 | 1143 | NBFNet [4] |
| WN V4 [1] | 9 | 3861 | 7940 | 3861 | 7940 | 934 | 7084 | 12334 | 2823 | A*Net [21] |
| NELL V1 [1] | 14 | 3103 | 4687 | 3103 | 4687 | 414 | 225 | 833 | 201 | RED-GNN [3] |
| NELL V2 [1] | 88 | 2564 | 8219 | 2564 | 8219 | 922 | 2086 | 4586 | 935 | RED-GNN [3] |
| NELL V3 [1] | 142 | 4647 | 16393 | 4647 | 16393 | 1851 | 3566 | 8048 | 1620 | RED-GNN [3] |
| NELL V4 [1] | 76 | 2092 | 7546 | 2092 | 7546 | 876 | 2795 | 7073 | 1447 | RED-GNN [3] |
| ILPC-small [52] | 48 | 10230 | 78616 | 6553 | 29060 | 2908 | 6653 | 29060 | 2902 | NodePiece [19] |
| ILPC-large [52] | 65 | 46626 | 202446 | 29246 | 77044 | 10179 | 29246 | 77044 | 10184 | NodePiece [19] |

Table 9: Statistics of fully-inductive datasets.

| Datasets | Training graphs | | | Validation graphs | | | | Test graphs | | | | SOTA |
|---|---|---|---|---|---|---|---|---|---|---|---|---|
| | #Ent. | #Rel. | #Facts | #Ent. | #Rel. | #Facts | #Valid. | #Ent. | #Rel. | #Facts | #Test | |
| FB-25 [5] | 5190 | 163 | 91571 | 4097 | 216 | 17147 | 5716 | 4097 | 216 | 17147 | 5716 | InGram [5] |
| FB-50 [5] | 5190 | 153 | 85375 | 4445 | 205 | 11636 | 3879 | 4445 | 205 | 11636 | 3879 | InGram [5] |
| FB-75 [5] | 4659 | 134 | 62809 | 2792 | 186 | 9316 | 3106 | 2792 | 186 | 9316 | 3106 | InGram [5] |
| FB-100 [5] | 4659 | 134 | 62809 | 2624 | 77 | 6987 | 2329 | 2624 | 77 | 6987 | 2329 | InGram [5] |
| WK-25 [5] | 12659 | 47 | 41873 | 3228 | 74 | 3391 | 1130 | 3228 | 74 | 3391 | 1131 | InGram [5] |
| WK-50 [5] | 12022 | 72 | 82481 | 9328 | 93 | 9672 | 3224 | 9328 | 93 | 9672 | 3225 | InGram [5] |
| WK-75 [5] | 6853 | 52 | 28741 | 2722 | 65 | 3430 | 1143 | 2722 | 65 | 3430 | 1144 | InGram [5] |
| WK-100 [5] | 9784 | 67 | 49875 | 12136 | 37 | 13487 | 4496 | 12136 | 37 | 13487 | 4496 | InGram [5] |
| NL-0 [5] | 1814 | 134 | 7796 | 2026 | 112 | 2287 | 763 | 2026 | 112 | 2287 | 763 | InGram [5] |
| NL-25 [5] | 4396 | 106 | 17578 | 2146 | 120 | 2230 | 743 | 2146 | 120 | 2230 | 744 | InGram [5] |
| NL-50 [5] | 4396 | 106 | 17578 | 2335 | 119 | 2576 | 859 | 2335 | 119 | 2576 | 859 | InGram [5] |
| NL-75 [5] | 2607 | 96 | 11058 | 1578 | 116 | 1818 | 606 | 1578 | 116 | 1818 | 607 | InGram [5] |
| NL-100 [5] | 1258 | 55 | 7832 | 1709 | 53 | 2378 | 793 | 1709 | 53 | 2378 | 793 | InGram [5] |

effectively learn representations and achieve high performance. Second, most existing KG reasoning models are developed based on several transductive datasets such as FB15k-237 [53], WN18RR [12], YAGO3-10 [55], and NELL-995 [54]. Models specifically designed for these datasets also contribute to high performance. Nevertheless, our KG-ICL still achieves results superior to supervised baseline models on 11 transductive datasets.

A few extra inductive datasets [15, 79] and fully-inductive data-sets [26] are often evaluated under the 1 vs. 50 setting, where the target entity is selected from 50 randomly sampled candidates. In [6], it evaluates on them under the full candidate setting, which reduces the uncertainty caused by random samples and provides a more stable evaluation. Therefore, we also conduct comparative experiments with ULTRA on these datasets under the full candidate setting. The results are reported in Table 12. We observe that KG-ICL outperforms ULTRA on most datasets.

# H   Limitations

The evaluations on 43 datasets demonstrate the proposed in-context KG foundation model's performance and generalization across transductive and inductive settings. Nonetheless, there are several limitations and open questions. Some KGs have special facts in addition to the mainstream triple facts involving subject, object entities, and their relations. These include facts with time stamps and facts containing multi-relational aspects. The foundation model for these special KGs also deserves attention. Scalability is an open challenge faced by existing KG reasoning models. Our proposed model addresses this by extracting a few prompt graphs with a small scale to represent relations, which has been demonstrated as a scalable approach in Appendix E.2. Our evaluations on large-scale datasets containing millions of facts also confirm its scalability. In future work, we plan to further enhance the scalability by incorporating strategies such as pruning and parallelization.

Table 10: Statistics of transductive datasets.

| Datasets | #Ent. | #Rel. | #Training | #Valid. | #Test | SOTA |
|---|---|---|---|---|---|---|
| CoDEx-small [56] | 2034 | 42 | 32888 | 1827 | 1828 | ComplEx RP [80] |
| WDsinger [57] | 10282 | 35 | 16442 | 1641 | 1640 | LR-GCN [81] |
| FB15k-237-10 [57] | 11512 | 237 | 27211 | 15624 | 18150 | DacKGR [82] |
| FB15k-237-20 [57] | 13166 | 237 | 54423 | 16963 | 19776 | DacKGR [82] |
| FB15k-237-50 [57] | 14149 | 237 | 136057 | 17449 | 20324 | DacKGR [82] |
| FB15k-237 [53] | 14541 | 237 | 272115 | 17535 | 20466 | NBFNet [4] |
| CoDEx-medium [56] | 17050 | 69 | 185584 | 10390 | 10391 | ComplEx RP [80] |
| NELL23k [57] | 22925 | 200 | 25445 | 4961 | 4952 | LR-GCN [81] |
| WN18RR [12] | 40943 | 11 | 86835 | 3034 | 3134 | NBFNet [4] |
| AristoV4 [58] | 44949 | 1605 | 242567 | 20000 | 20000 | ComplEx RP [80] |
| Hetionet [61] | 45158 | 24 | 2025177 | 112510 | 112510 | RotatE [14] |
| NELL-995 [54] | 74536 | 200 | 149678 | 543 | 2818 | RED-GNN [3] |
| CoDEx-large [56] | 77951 | 69 | 551193 | 30622 | 30622 | ComplEx RP [80] |
| ConceptNet100k [60] | 78334 | 34 | 100000 | 1200 | 1200 | BiQUE [83] |
| DBpedia100k [59] | 99604 | 470 | 597572 | 50000 | 50000 | ComplEx-NNE+AER [84] |
| YAGO3-10 [55] | 123182 | 37 | 1079040 | 5000 | 5000 | NBFNet [4] |

Table 11: Detailed results on 43 datasets.

| Datasets | Supervised SOTA | | ULTRA pre-train | | KG-ICL pre-train | | ULTRA finetune | | KG-ICL finetune | |
|---|---|---|---|---|---|---|---|---|---|---|
| | MRR | H@10 | MRR | H@10 | MRR | H@10 | MRR | H@10 | MRR | H@10 |
| FB V1 | 0.457 | 0.589 | 0.498 | 0.656 | 0.520 | 0.678 | 0.509 | 0.670 | **0.531** | **0.700** |
| FB V2 | 0.510 | 0.672 | 0.512 | 0.700 | 0.565 | 0.749 | 0.524 | 0.710 | **0.568** | **0.768** |
| FB V3 | 0.476 | 0.637 | 0.491 | 0.654 | 0.535 | 0.695 | 0.504 | 0.663 | **0.537** | **0.704** |
| FB V4 | 0.466 | 0.645 | 0.486 | 0.677 | 0.513 | 0.699 | 0.496 | 0.684 | **0.525** | **0.706** |
| ILPC-large | 0.070 | 0.146 | 0.290 | 0.424 | 0.288 | 0.412 | **0.308** | **0.431** | 0.295 | 0.411 |
| ILPC-small | 0.130 | 0.251 | 0.302 | 0.443 | 0.288 | 0.446 | 0.303 | 0.453 | **0.316** | **0.473** |
| NELL V1 | 0.637 | 0.866 | 0.785 | 0.913 | 0.693 | 0.915 | 0.757 | 0.878 | **0.841** | **0.995** |
| NELL V2 | 0.419 | 0.601 | 0.526 | 0.707 | **0.644** | **0.835** | 0.575 | 0.761 | 0.641 | 0.835 |
| NELL V3 | 0.436 | 0.594 | 0.515 | 0.702 | 0.613 | 0.792 | 0.563 | 0.755 | **0.631** | **0.799** |
| NELL V4 | 0.363 | 0.556 | 0.479 | 0.712 | 0.590 | 0.791 | 0.469 | 0.733 | **0.594** | **0.802** |
| WN V1 | 0.741 | 0.826 | 0.648 | 0.768 | 0.733 | **0.838** | 0.685 | 0.793 | **0.762** | 0.827 |
| WN V2 | 0.704 | **0.798** | 0.663 | 0.765 | 0.696 | 0.783 | 0.679 | 0.779 | **0.721** | 0.787 |
| WN V3 | 0.452 | 0.568 | 0.376 | 0.476 | 0.425 | 0.548 | 0.411 | 0.546 | **0.503** | **0.626** |
| WN V4 | 0.661 | 0.743 | 0.611 | 0.705 | 0.652 | 0.722 | 0.614 | 0.720 | **0.683** | **0.749** |
| FB-25 | 0.223 | 0.371 | 0.388 | 0.640 | 0.396 | 0.656 | 0.383 | 0.635 | **0.434** | **0.694** |
| FB-50 | 0.189 | 0.325 | 0.338 | 0.543 | 0.341 | 0.559 | 0.334 | 0.538 | **0.384** | **0.598** |
| FB-75 | 0.117 | 0.218 | 0.403 | 0.604 | 0.438 | 0.633 | 0.400 | 0.598 | **0.458** | **0.664** |
| FB-100 | 0.133 | 0.271 | 0.449 | 0.642 | 0.487 | 0.694 | 0.444 | 0.643 | **0.499** | **0.703** |
| NL-0 | 0.309 | 0.506 | 0.342 | 0.523 | **0.557** | **0.777** | 0.329 | 0.551 | 0.555 | 0.765 |
| NL-25 | 0.261 | 0.464 | 0.395 | 0.569 | **0.550** | **0.736** | 0.407 | 0.596 | 0.540 | 0.730 |
| NL-50 | 0.281 | 0.453 | 0.407 | 0.570 | **0.534** | 0.704 | 0.418 | 0.595 | 0.528 | **0.708** |
| NL-75 | 0.334 | 0.501 | 0.368 | 0.547 | **0.452** | 0.673 | 0.374 | 0.570 | 0.446 | **0.681** |
| NL-100 | 0.269 | 0.431 | 0.471 | 0.651 | 0.556 | 0.762 | 0.458 | 0.684 | **0.557** | **0.766** |
| WK-25 | 0.107 | 0.169 | 0.316 | 0.532 | 0.423 | 0.621 | 0.321 | 0.535 | **0.425** | **0.628** |
| WK-50 | 0.247 | 0.362 | 0.166 | 0.324 | 0.273 | 0.430 | 0.140 | 0.280 | **0.277** | **0.432** |
| WK-75 | 0.068 | 0.135 | 0.365 | 0.537 | 0.437 | 0.602 | 0.380 | 0.530 | **0.466** | **0.626** |
| WK-100 | 0.186 | 0.309 | 0.164 | 0.286 | 0.262 | 0.409 | 0.168 | 0.286 | **0.270** | **0.415** |
| AristoV4 | 0.311 | 0.447 | 0.182 | 0.282 | 0.203 | 0.306 | **0.343** | **0.496** | 0.313 | 0.480 |
| CoDEx-small | 0.473 | 0.663 | 0.472 | 0.667 | 0.465 | 0.654 | **0.490** | **0.686** | 0.479 | 0.662 |
| CoDEx-medium | 0.352 | 0.490 | 0.372 | 0.525 | 0.330 | 0.474 | 0.372 | 0.525 | **0.402** | **0.565** |
| CoDEx-large | 0.345 | 0.473 | 0.338 | 0.469 | 0.261 | 0.376 | 0.343 | 0.478 | **0.388** | **0.508** |
| ConceptNet100K | 0.320 | 0.553 | 0.082 | 0.162 | 0.249 | 0.416 | 0.310 | 0.529 | **0.371** | **0.584** |
| DBpedia100K | 0.306 | 0.418 | 0.398 | 0.576 | 0.390 | 0.541 | 0.436 | 0.603 | **0.455** | **0.604** |
| FB15k-237 | **0.415** | **0.599** | 0.368 | 0.564 | 0.359 | 0.541 | 0.368 | 0.564 | 0.376 | 0.538 |
| FB15k-237-10 | 0.219 | 0.337 | 0.248 | 0.398 | **0.274** | **0.433** | 0.254 | 0.411 | 0.260 | 0.416 |
| FB15k-237-20 | 0.247 | 0.391 | 0.272 | 0.436 | **0.285** | **0.454** | 0.274 | 0.445 | 0.284 | 0.456 |
| FB15k-237-50 | 0.293 | 0.458 | 0.324 | 0.526 | **0.329** | 0.520 | 0.325 | **0.528** | 0.324 | 0.499 |
| Hetionet | 0.257 | 0.403 | 0.257 | 0.379 | 0.260 | 0.371 | **0.399** | **0.538** | 0.269 | 0.402 |
| NELL-995 | **0.543** | 0.651 | 0.406 | 0.543 | 0.532 | 0.653 | 0.509 | 0.660 | 0.534 | **0.672** |
| NELL23K | 0.253 | 0.419 | 0.239 | 0.408 | 0.317 | 0.532 | 0.268 | 0.450 | **0.329** | **0.552** |
| WD-singer | 0.393 | 0.500 | 0.382 | 0.498 | 0.470 | 0.582 | 0.417 | 0.526 | **0.493** | **0.599** |
| WN18RR | **0.551** | **0.666** | 0.480 | 0.614 | 0.455 | 0.527 | 0.480 | 0.614 | 0.536 | 0.637 |
| YAGO3-10 | **0.563** | 0.708 | 0.451 | 0.615 | 0.352 | 0.503 | 0.557 | **0.710** | 0.545 | 0.688 |
| Average | 0.351 | 0.493 | 0.396 | 0.557 | 0.442 | 0.606 | 0.421 | 0.590 | **0.473** | **0.638** |

Table 12: Results on more datasets under the full candidate setting.

| Datasets | ULTRA pre-train | | KG-ICL pre-train | | ULTRA finetune | | KG-ICL finetune | |
|---|---|---|---|---|---|---|---|---|
| | MRR | H@10 | MRR | H@10 | MRR | H@10 | MRR | H@10 |
| HM 1k | 0.059 | 0.092 | 0.059 | 0.107 | 0.042 | 0.100 | **0.089** | **0.144** |
| HM 3k | 0.037 | 0.077 | 0.049 | 0.099 | 0.030 | 0.090 | **0.081** | **0.129** |
| HM 5k | 0.034 | 0.071 | 0.043 | 0.091 | 0.025 | 0.068 | **0.070** | **0.108** |
| IndigoBM | **0.440** | **0.648** | 0.351 | 0.558 | 0.432 | 0.639 | **0.440** | 0.641 |
| MT1 tax | 0.224 | 0.305 | 0.280 | 0.451 | 0.330 | 0.459 | **0.411** | **0.521** |
| MT1 health | 0.298 | 0.374 | 0.378 | 0.464 | 0.380 | 0.467 | **0.387** | **0.479** |
| MT2 org | 0.095 | 0.159 | 0.093 | 0.156 | **0.104** | 0.170 | 0.100 | **0.171** |
| MT2 sci | 0.258 | 0.354 | **0.326** | **0.476** | 0.311 | 0.451 | 0.303 | 0.396 |
| MT3 art | 0.259 | 0.402 | 0.258 | 0.406 | **0.306** | **0.473** | **0.306** | 0.460 |
| MT3 infra | 0.619 | 0.755 | 0.633 | 0.777 | 0.657 | 0.807 | **0.676** | **0.808** |
| MT4 sci | 0.274 | 0.449 | 0.296 | 0.470 | 0.303 | **0.478** | **0.307** | 0.473 |
| MT4 health | 0.624 | 0.737 | 0.648 | 0.767 | 0.704 | **0.785** | **0.710** | 0.776 |
| Metafam | 0.238 | 0.644 | 0.500 | 0.886 | 0.997 | **1.000** | **1.000** | **1.000** |
| FBNELL | 0.485 | 0.652 | 0.509 | 0.692 | 0.481 | 0.661 | **0.516** | **0.699** |

# I  Broader Impacts

Our work seeks to build a KG foundation model with effective, efficient, and transferable reasoning capabilities over unseen entities, relations, and even previously unseen KGs, all without requiring retraining from scratch. We believe that the proposed model has the potential to be applied in broad knowledge-driven applications, such as question-answering and recommender systems. Its ability to adapt to changes in the graph and generalize to unseen data will be beneficial in addressing issues such as cold start. Nevertheless, excessive reliance on knowledge from pre-training data and a few examples may lead to societal biases and unfairness. We have discussed the quality and potential impacts of the pre-training data in Appendix E.1. In practical applications, we also should carefully design example selection strategies to avoid potential societal biases and unfairness.

