# OpenReview forum: "A Prompt-Based Knowledge Graph Foundation Model for Universal In-Context Reasoning"
_NeurIPS.cc/2024/Conference — NeurIPS 2024 poster_

### Official Review · Reviewer_JyNi · 2024-07-08

**Soundness:** 3
**Presentation:** 3
**Contribution:** 2
**Rating:** 6
**Confidence:** 4

**Summary:**

This paper introduces KG-ICL, a model that facilitates generalized reasoning over knowledge graphs via in-context learning. KG-ICL first extracts example facts relevant to the query from the knowledge graph to generate prompt graphs. These prompt graphs are then encoded using a unified tokenizer and message passing neural network to produce relation representations. Next, the generated embeddings of relation entities are then integrated with knowledge graph to generate the score of candidate entities. Extensive experiments on 43 different knowledge graphs under transduction and inductive validate the effectiveness of KG-ICL.

**Strengths:**

1. The paper is well-structured.
2. Extensive experimental results demonstrate that KG-ICL can consistently outperform  baseline models.
3. Codes are provided for reproducibility.

**Weaknesses:**

1. The motivation for the ICL setting over KGs is not clearly motivation. It appears to be essentially the same as the inductive learning setting. Furthermore, the results shown in Figure 3 indicate that the number of examples has no significant effect on performance, which contradicts the hypothesis of ICL.
2. In the section 4.1, the authors propose to sample $M$ example and extract the prompt graph within $k$ hop paths. However, it's unclear whether the number of $M$ and $k$ will affect the model performance. Unfortunately, the authors provide little explanation for this.
3. The paper introduces the unified tokenizer as a component of the KG-ICL model, yet the motivation for its use is not clear. Additionally, the ablation studies (Table 2) show that excluding the unified tokenizer or the token representation results in only a marginal performance decline. This raises questions about the necessity and practical advantage of using the unified tokenizer for the reasoning task.

**Questions:**

1. How KG-ICL generate the prompt graph? Does it directly sample the example from the knowledge graph and then connect them?
2. It is ambiguous what is the difference between the w/o unified tokenizer and w/o token represent.
3. In this paper, the task is assuming that a model is pre-trained using a set of source KGs. Is this a standard task setting? How realistic is this setting for solving some real problem?

**Limitations:**

The authors have discussed the limitations of their work in the paper.

---

> ### Author Rebuttal · Authors · 2024-08-06
>
> Dear Reviewer JyNi,
>
> We deeply appreciate your valuable comments and dedication during the review period. We sincerely hope that our response can ease all your concerns. Please feel free to contact us with any further comments or require additional clarification.
>
> **W1: Details about in-context setting.**
>
> * **Differences between in-context and inductive settings.** The inductive setting is defined as only reasoning on a single KG, whereas the in-context setting uses a universal model for diverse KGs. The in-context setting not only covers both transductive and inductive settings but can also be generalized to unseen KGs.
>
> * **Motivation for in-context setting.** This work aims at a foundation model applicable to diverse KGs. To generalize the model to unseen KGs, we draw inspiration from in-context learning in graph and language modeling to propose the in-context KG reasoning setting. This setting allows one model to generalize to diverse KGs based on just a few examples without updating parameters.
>
> * **Results are not sensitive to $M$.** We discuss reasons for this in Lines 334-338. In in-context learning, adding more examples does not always result in significant performance gains. Additional examples might be redundant or even introduce noise, and many queries may not be difficult, so a few examples suffice. Moreover, effective prompt engineering also results in high efficiency of example utilization, one of our strengths.
>
> **W2: Impact of the number of $M$ and $k$.**
>
> We believe you missed these results in the submission. Section 5.3 analyzes the impact of $M$ and $k$.
>
> * $M$: Figure 3 reports the MRR results with different numbers of $M$. KG-ICL can unleash universal reasoning capabilities with only a few examples.
> * $k$: Table 3 shows the results of using diverse $k$-hop paths. The “3-hop path” variant performs well, but the “1-hop” and “2-hop” variants are not effective enough for reasoning. We do not use higher values of $k$ due to the high cost.
>
> **W3: Details about unified tokenizer.**
>
> * **Motivation for unified tokenizer.** Different KGs contain distinct entities and relations, which is the key challenge for generalizing to unseen KGs. Previous studies learn a specific embedding for each entity and relation, and rely on GNNs to capture the relational structures for knowledge transfer. In our work, we find that transferable entity and relation embeddings can also facilitate knowledge transfer because the inductive bias of entities and relations also helps generalize unseen data. Motivated by this, we propose the unified tokenizer, which maps entities and relations from various KGs to a shared token list based on their relative positions in the prompt graphs. Our experiments show that the proposed unified tokenizer can help knowledge transfer and outperform previous studies.
>
> * **Why does it still work when removing it?** After removing it, we use randomly re-initialized features as input embeddings of entities and relations in prompt graphs during training and testing. It still works because i) previous studies [1, 2, 3] show that GNNs can work with random node initialization. InGram [1] uses randomly re-initialized features and performs relatively well for KG reasoning. ii) Other modules of KG-ICL can also support knowledge transfer, e.g., the results of the “w/o prompt graph” variant demonstrate that the KG encoder alone can also learn to reason.
>
> Although it still works, the MRR scores significantly decrease from 0.442 to 0.403 without the assistance of the unified tokenizer, demonstrating its impact. Please see Q2 for the difference between ablation variants.
>
> [1] InGram: Inductive knowledge graph embedding via relation graphs. ICML 2023
>
> [2] The surprising power of graph neural networks with random node initialization. IJCAI 2021
>
> [3] Random features strengthen graph neural networks. ICDM 2021
>
> **Q1: How to generate the prompt graph.**
>
> Section 4.1 describes the prompt graph generation process. We generate a prompt graph for each example fact rather than directly connecting them.
>
> * Given a query relation, we first sample $M$ example facts as Eq. (1).
> * For each example fact, we include the entities in the $k$-hop paths between the subject and object entities, along with the 1-hop neighbors of the subject and object entities, in its prompt graph, as in Eq. (2).
> * Finally, we extract the facts and relations among the above entities (Lines 157-159).
>
> After the generation, we encode each prompt graph and use mean-pooling to obtain the final prompts, as in Eq. (6).
>
> **Q2: Differences between the “w/o unified tokenizer” and “w/o token represent”.**
>
> The “w/o unified tokenizer” variant uses randomly re-initialized vectors as entity and relation embeddings during training and testing. The “w/o token represent.” variant keeps the mapping of entities and relations to unified tokens but replaces learnable token embeddings with non-learnable one-hot labeling vectors from GraIL [4]. For clarity, we will rename “w/o token represent.” to “w/ GraIL's one-hot labeling”.
>
> [4]  Inductive relation prediction by subgraph reasoning. ICML 2020
>
> **Q3: Is pre-training using a set of source KGs a standard setting? How realistic is this setting for solving real problems?**
>
> Yes. It is a standard setting for the KG foundation model pre-training [5].
>
> This setting is realistic for solving real problems. In the real-world scenario, there are many open-source KGs [6]. Previous methods must train a separate model for each KG and struggle with handling the updates to the KG or unseen KGs. In contrast, our foundation model is only pre-trained once on a few open-source datasets and can be directly applied to various continuously updated KGs and unseen KGs. Our foundation model may also be directly applied to private KGs, such as companies’ product graphs or personal healthcare KGs.
>
> [5] Towards foundation models for knowledge graph reasoning. ICLR 2024
>
> [6] https://lod-cloud.net/

---

> > ### Comment · Reviewer_JyNi · 2024-08-09
> > **Thanks for the response**
> >
> > Thanks for the response. My concerns are well addressed. I will increase my score to 6.

---

> > > ### Author Response · Authors · 2024-08-09
> > > **Grateful Thanks to Reviewer JyNi**
> > >
> > > Dear Reviewer JyNi,
> > >
> > > We sincerely appreciate your recognition of our efforts to address your concerns. Thank you for taking the time to provide insightful suggestions that will help us further refine our paper.
> > >
> > > Best regards,
> > >
> > > Authors

---

### Official Review · Reviewer_4T5d · 2024-07-09

**Soundness:** 3
**Presentation:** 4
**Contribution:** 3
**Rating:** 7
**Confidence:** 5

**Summary:**

The paper studies inductive KG reasoning with unseen entities and relations at inference time and introduces KG-ICL, an in-context learning model for KG completion. Following Ultra [1], KG-ICL employs a two-stage approach: (1) obtaining relational representations based on the given graph; (2) performing entity prediction on the given graph using those relational representations.

The main difference with ULTRA is Stage 1, that is, instead of creating a graph of relations and learning tokens for meta-relations, KG-ICL mines several (around 5) examples of k-hop subgraphs around each relation type and applies a different labeling trick learning a different vocabulary of transferable tokens. Nodes are labeled following the distance encoding of GraIL [2] (tokens represent all combinations of pairwise distances up to hop $k$), relations are labeled simply by a binary indicator whether a given relation is the query relation or not (two tokens). Applying a GNN over each subgraph, their final representations are mean pooled to get a singe tensor of relational representations.

In Stage 2, those representations are used in a standard inductive pipeline for initializing edge type vectors, putting the query relation vector on the starting head node, and running another GNN to get the final predictions. Experimentally, KG-ICL demonstrates promising results and outperforms Ultra on inductive datasets while being marginally better on larger transductive datasets.

**Strengths:**

**S1.** Foundation models for inductive KG reasoning on any unseen graph is a timely and important topic. New non-trivial approaches in this field are rare and this paper does a good job presenting and explaining the details clearly.

**S2.** A different approach for obtaining relational representations through few-shot subgraph examples of each relation (resembling the “in-context learning” scenario) and learning a transferable vocabulary - here it is an extended GraIL-based labeling with distance encoding for nodes and binary same/not-same indicator for relations (although Table 2 shows that learning tokens does not bring much benefits and labeling strategy is more important).

**S3**. Informative ablations, experiments on the Ultra benchmark involving 57 datasets

**Weaknesses:**

**W1.** The main problem of KG-ICL is a different pre-training dataset mixture (inductive FB V1, inductive NELL V1, transductive Codex Small) which makes it hard to directly compare the performance numbers against Ultra (that used only transductive FB15k237, WN18RR, and Codex Medium). For a fair comparison, KG-ICL should have been either pre-trained on the same datasets as Ultra, or Ultra should have been re-trained on the new mixture. The new pre-training mixture consists of smaller graphs and is better suited for inference on smaller inductive datasets (where KG-ICL gains are the largest). While it is possible to pull the argument that LLMs and FMs are trained on different datasets (often undisclosed to the public) and evaluated on the same benchmarks, I believe the Ultra benchmarking datasets for KG reasoning are open and transparent enough for conducting fair evaluations.

**W2** Since KG-ICL focuses on the zero-shot inference performance (line 239), one missing experiment is to measure the performance as a function of training graphs in the pre-training mixture - for instance, Ultra provides several checkpoints with the growing inductive inference performance with more datasets added to the training.

References:
[1] Galkin et al. Towards Foundation Models for Knowledge Graph Reasoning. ICLR 2024.
[2] Teru et al. Inductive relation prediction by subgraph reasoning. ICML 2020.
[3] Huang et al. A Theory of Link Prediction via Relational Weisfeiler-Leman on Knowledge Graphs. NeurIPS 2023.

**Questions:**

**Q1.** Why were 4 inductive (HM) and 10 fully-inductive (ISDEA) datasets omitted from the main results table? Those are all viable datasets and I don’t see a major reason to present the average over 43 datasets instead of 57.

**Q2.** Line 168: the formula for the total number of tokens $\frac{(k+1)(k+2)}{2} - 2$ seems to be incorrect? Setting $k=2$, the formula gives 4 tokens but we only have $(0,1), (1,0), (1,1)$ distances. Similarly, for $k=4$, there are 15 combinations of distances (without (0,0)) but the formula gives 13 options

**C1**. Line 190: citations on conditional message passing GNNs miss [3] that theoretically formalized C-MPNNs.

**Limitations:**

Addressed

---

> ### Author Rebuttal · Authors · 2024-08-06
>
> Dear Reviewer 4T5d,
>
> We sincerely appreciate your invaluable time and positive comments. Your insightful suggestions give us a great opportunity to improve our paper. We conduct extra experiments and provide further analyses, which we will incorporate into the paper. We sincerely hope these enhancements meet your expectations and contribute to the overall quality of our work.
>
> **W1: Use the same pre-training dataset mixture with ULTRA.**
>
> According to your suggestion, we use the same pre-training dataset mixture with ULTRA for a fair comparison. The results are shown below:
>
> ||Induct.|Full-induct.|Transd.|Avg|
> |:---:|:---:|:---:|:---:|:---:|
> ||MRR/H@10|MRR/H@10|MRR/H@10|MRR/H@10|
> |ULTRA (FB15k-237 WN18RR CoDEx-medium)|.513/.664|.352/.536|.329/.479|.396/.557|
> |KG-ICL (FB15k-237 WN18RR CoDEx-medium)|.547/.700|.431/.629|**.357/.506**|.441/**.606**|
> |KG-ICL (FB V1 NELL V1 CoDEx-small)|**.554/.707**|**.439/.635**|.346/.493|.**442/.606**|
>
> We observe that KG-ICL still outperforms ULTRA in this setting. Pre-training with this dataset mixture causes a slight decrease in the inductive performance and a slight improvement in the transductive results, but neither change is significant.
>
> We use smaller datasets because i) we find that using smaller datasets does not significantly impact model performance. ii) ULTRA is pre-trained with two A100 40GB GPUs, while we pre-train using a single 3090 24GB GPU. We find that smaller pre-training datasets allow a higher batch size, resulting in more stable pre-training. In this rebuttal, we pre-train KG-ICL with an A6000 48GB GPU to handle the new dataset mixture.
>
> We sincerely thank you for this insightful suggestion. We will include these results and analyses in the revision.
>
> **W2: Pre-training with more datasets.**
>
> We conduct experiments on growing pre-training mixtures, sequentially adding pre-training datasets in the same order as in Table 9 and Figure 6 of [6], i.e., FB15k-237, WN18RR, CoDEx-medium, NELL-995, YAGO3-10, ConceptNet100K, DBpedia100K, and AristoV4.
> The results are shown in the **PDF file of the Author Rebuttal for all reviewers**. We will include the results in the revision.
>
> We observe that the performance improves with the number of pre-training datasets. Unlike ULTRA, KG-ICL even performs well with pre-training on a single KG. This improvement is due to two key factors: first, we generate a diverse set of prompt graphs for different relations within the same KG, which increases sample diversity. Second, our targeted prompt engineering reduces learning complexity and facilitates better generalization.
>
> **Q1: Why are some datasets not included in the main results table?**
>
> We discuss the reason for this in Lines 767-772. The results in the main table are evaluated in the full candidate setting, and all entities are considered as candidates. However, the results of the supervised SOTA models on these datasets, e.g., INDIGO [78] on HM 1k [15], are evaluated in the 50-negative setting. Besides, ULTRA [6] also does not include these datasets in the total average result calculation (as the main results in Table 1) and presents these results in a separate table (Table 11). Therefore, we report the results in a separate table for clarity.
>
> **Q2: Number of tokens.**
>
> The correct number of tokens is $\frac{(k+1)\times(k+2)}{2}-2(k-1)$. We will clarify this in the revision. Thank you for the heads up.
>
> For an entity's position $(i, j)$, $i$ and $j$ represent its shortest path lengths to the example subject and object entities, respectively. Due to the condition $i+j\leq k$, the entity positions form a triangular region with $\frac{(k+1)\times(k+2)}{2}$ tokens.
>
> This token set can be further optimized, as some tokens are not used.
> For example k=2:
>
> 1 2 x
>
> 3 4
>
> x
>
> In the above example, “x” denotes the unused tokens and the numbers denote the used tokes. Note that
> * We keep the token at position (0, 0) because of the existence of the edges where the subject and object entities are the same, e.g., (league_nfl, competesWith, league_nfl) from NELL-995.
> * The tokens in positions (0, 2) and (2, 0) are not used because $i=0$ and $j=0$ indicate the example subject and object entity, respectively. There is an example fact (edge) between them so the shortest path length between them is less or equal to 1.
>
> For another example k=4:
>
> 1 2 x x x
>
> 3 4 5 6
>
> x 7 8
>
> x 9
>
> x
>
> Similar to the case with $k=2$, there are $2\times (4-1)$ tokens not used due to the distance between the example subject and object is less or equal to 1. Therefore, the total number of tokens used is $\frac{(k+1)\times(k+2)}{2}-2(k-1)$.
>
> We use $(k+1) \times (k+1)$ token embeddings in the source code for convenience. The unused placeholder token embeddings do not participate in training or testing and thus do not affect the experimental results.
>
> **C1: Citation.**
>
> We appreciate you providing this citation, which theoretically formalizes C-MPNNs. We will add this citation in Line 190 and the related work section.

---

> > ### Comment · Reviewer_4T5d · 2024-08-08
> > **Response**
> >
> > Thank you for the clarifications and new experimental results, I am pretty satisfied with the response and increasing the score to 7.

---

> > > ### Author Response · Authors · 2024-08-08
> > > **Appreciation to Reviewer 4T5d**
> > >
> > > Dear Reviewer 4T5d,
> > >
> > > We are grateful for your prompt response and for increasing your score. Thank you for your recognition and support of our work.
> > >
> > > Best regards,
> > >
> > > Authors

---

### Official Review · Reviewer_xJpJ · 2024-07-12

**Soundness:** 3
**Presentation:** 3
**Contribution:** 2
**Rating:** 7
**Confidence:** 3

**Summary:**

This paper aims to build a foundation model for knowledge graphs, to have a universal reasoning ability across diverse knowledge graphs including the unseen entities and relations. Specifically, given the query, the proposed approach first extracts its relevant prompt graphs and then map their entities and relations to the predefined tokens (versatile for any entities and relations). After that, it performs two message passing, in order to encode the prompt graphs and perform knowledge graph reasoning with them. The authors extensively validate the proposed approach on 43 knowledge graphs, showcasing its effectiveness over the strong knowledge graph foundational model baseline.

**Strengths:**

* Building a foundation model for knowledge graphs is a very important task.
* The proposed (in-context learning style) approach to extract prompt graphs (for the given query) and use them to perform knowledge graph reasoning is reasonable.
* The proposed approach outperforms the previous knowledge graph foundation model by large margins.
* This paper is well-written.

**Weaknesses:**

* The technical contribution of the proposed approach over the prior knowledge graph foundation work [6] looks marginal, which is also not clearly discussed in the paper. In my view, the major improvement (and the novelty) of the proposed approach is to extract multiple prompt graphs (relevant to the query) and use them for knowledge graph reasoning, instead of using only one target subgraph.
* The performance comparison between high and low resources relations (or entities) is worthwhile to present (in the main paper).

**Questions:**

Please see my weaknesses above.

**Limitations:**

The authors discuss the limitations and potential negative societal impact of their work.

---

> ### Author Rebuttal · Authors · 2024-08-06
>
> Dear Reviewer xJpJ,
>
> We sincerely thank you for your positive comments and valuable suggestions. If you have any further suggestions, please let us know. We would be happy to continue the discussion.
>
> **Q1: Key factor for performance improvement.**
>
> The proposed local context prompt graph for relation representation is the primary factor behind the significant performance improvement. It effectively distinguishes important relations from noisy ones and preserves the connectivity between entities.
>
> The results in Figure 3 show that even with just 1 or 3 prompt graphs, our model outperforms ULTRA [6], demonstrating that the number of graphs is not the key factor. Instead, it is the superior relation modeling of our model that makes the difference.
>
> ULTRA [6] often fails to distinguish between important and irrelevant relations due to its global relation graph. For example, as long as there exists at least one father who has a teacher and one teacher who has a father, there will be two edges "*teaches*->*fatherOf*" and "*fatherOf*->*teaches*" in ULTRA’s relation graph. This will mislead the model to consider “*teaches*” as an important relation for inferring the "*fatherOf*"-related queries.
>
> Instead, in our prompt graph with an example fact (A, *fatherOf*, B), since there is a path (A, *husbandOf*, C)->(C, *motherOf*, B) from A to B, it becomes clear that the relations such as "*motherOf*" are more important than the relations not included in any path. This better relation modeling reduces the impact of irrelevant context, thereby enhancing reasoning.
>
> **Q2: The performance comparison between high and low resources relations in Appendix E.4 is worth presenting in the main paper.**
>
> Thanks for your suggestion. This experiment shows the advantage of our model in modeling low-resource relations, which can be beneficial for future research in this field. We will move this experiment to the main paper in the revision.

---

> > ### Comment · Reviewer_xJpJ · 2024-08-12
> >
> > Thank you for your response, which addresses all of my concerns. This is a good paper and I increase the rating (from 6) to 7.

---

> > > ### Author Response · Authors · 2024-08-13
> > > **Appreciation to Reviewer xJpJ**
> > >
> > > Dear Reviewer xJpJ,
> > >
> > > We sincerely appreciate your support and for revising the rating. Thank you once again for your invaluable time and effort during the review and discussion periods.
> > >
> > > Best regards,
> > >
> > > The Authors

---

### Author Rebuttal · Authors · 2024-08-06

Dear Reviewers,

We express our sincere gratitude for your invaluable time and dedication throughout the review period. We sincerely appreciate your positive comments acknowledging that
* The foundation model for KG reasoning on any unseen graph is a timely and important topic. New non-trivial approaches in this field are rare, and this paper does a good job of presenting and explaining the details clearly.
* The proposed approach outperforms the previous model by large margins.
* Informative ablations, experiments on the benchmark involving 57 datasets.
* The paper is well-written and well-structured.
* Codes are provided for reproducibility.

This work aims at a universal foundation model applicable to diverse KGs. The contributions of this work are listed below:
* Our key contribution is an in-context KG reasoning foundation model, namely KG-ICL.
* We present three novel modules to achieve universal KG reasoning:
    * We present a local context prompt graph to outline the reasoning pattern for specific query relation, which achieves better relation modeling by incorporating example-edge-centered local subgraph contexts.
    * We employ a unified tokenizer to map entities and relations to shared tokens, facilitating knowledge transfer.
    * We also propose two message-passing neural networks for prompt encoding and KG reasoning.
* We conduct extensive experiments on various KGs in both transductive and inductive settings. Results indicate that KG-ICL outperforms baselines on most datasets, showcasing its outstanding generalization and universal reasoning capabilities.

We sincerely hope that our response has properly addressed all your concerns. We extend our heartfelt thanks for your insightful suggestions and positive comments and eagerly await your valuable feedback during the discussion period.

Best,

Authors

---

### Decision · Program_Chairs · 2024-09-25

**Decision:**

Accept (poster)

**Comment:**

This paper presents a new approach to in-context learning within knowledge graphs (KG), delving into the realm of inductive reasoning over knowledge graphs for previously unseen entities and relationships during inference. Its main contributions include building an in-context KG foundation model and strong empirical results.

All the main concerns raised by the reviewers, e.g., the need for ablation studies on crucial performance factors and the inclusion of further experiments for baseline comparison, were thoroughly addressed in the discussion phase.

To sum up, all reviewers are positive to the paper and agree that the paper makes a significant contribution. I agree with the reviewers and recommend its acceptance.